# CVGL: Causal Learning and Geometric Topology

**Songsong Ouyang**     **Yingying Zhu**[*]
College of Computer Science and Software Engineering
Shenzhen University
2400101027@mails.szu.edu.cn, zhuyy@szu.edu.cn

## Abstract

Cross-view geo-localization (CVGL) aims to estimate the geographic location of a street image by matching it with a corresponding aerial image. This is critical for autonomous navigation and mapping in complex real-world scenarios. However, the task remains challenging due to significant viewpoint differences and the influence of confounding factors. To tackle these issues, we propose the Causal Learning and Geometric Topology (CLGT) framework, which integrates two key components: a Causal Feature Extractor (CFE) that mitigates the influence of confounding factors by leveraging causal intervention to encourage the model to focus on stable, task-relevant semantics; and a Geometric Topology Fusion (GT Fusion) module that injects Bird's Eye View (BEV) road topology into street features to alleviate cross-view inconsistencies caused by extreme perspective changes. Additionally, we introduce a Data-Adaptive Pooling (DA Pooling) module to enhance the representation of semantically rich regions. Extensive experiments on CVUSA, CVACT, and their robustness-enhanced variants (CVUSA-C-ALL and CVACT-C-ALL) demonstrate that CLGT achieves state-of-the-art performance, particularly under challenging real-world corruptions. Our codes are available at CLGT.

## 1 Introduction

Cross-view geo-localization (CVGL) aims to estimate the geographic location of a street image by matching it to a corresponding aerial image. This task plays a crucial role in applications such as autonomous driving, robotic navigation, and urban mapping [24; 2; 27]. However, it remains highly challenging due to the extreme differences in perspective, scale, appearance, confounders and occlusion between street and aerial views. Previous studies have mainly explored three directions to improve cross-view matching: viewpoint modeling [28], spatial alignment [21], and hard negative mining [1]. Despite these efforts, cross-view

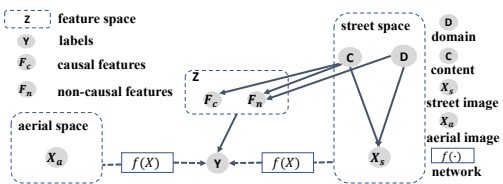

Figure 1: Structural Causal Model (SCM) for cross-view geo-localization. Nodes represent variables and arrows denote dependencies.

geo-localization remains challenging due to weather changes, misalignment, and occlusions, all of which demand stronger generalization and discriminative feature learning. To address these limitations, Mi et al.[13] introduced feature consistency constraints to enhance robustness to orientation and field-of-view variations. To better reflect real-world conditions, Zhang et al.[32] proposed corruption-rich benchmarks for robust evaluation, while Ye et al.[28] leveraged a Bird's Eye View (BEV) representation as an intermediate domain to bridge the large cross-view gap.

---

[*]Corresponding author

39th Conference on Neural Information Processing Systems (NeurIPS 2025).

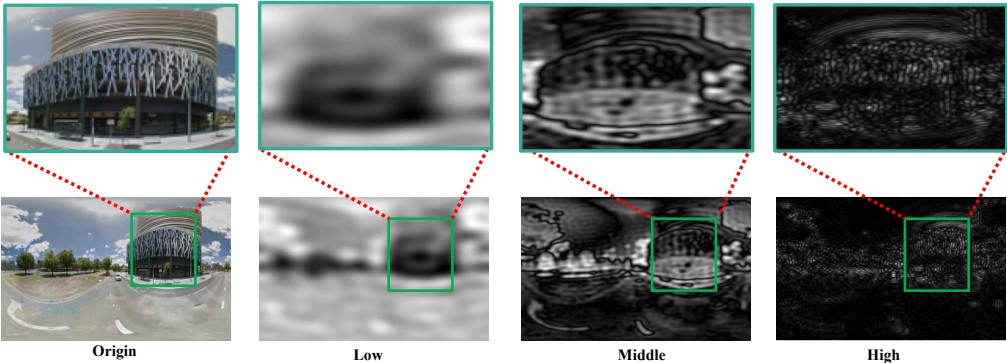

**Origin**   **Low**   **Middle**   **High**

Figure 2: Visualization of low, mid, and high frequency components of a street image. Low and high frequencies emphasize domain-specific information such as style, while the mid-frequency band retains domain-invariant cues such as structure and shape.

Building upon the aforementioned challenges and recent advances, we propose a framework to address cross-view geo-localization from both causal perspectives and geometric. Inspired by instances where classification models mistakenly associate sheep with grass, we argue that the CVGL task should not rely on confounding factors such as background or lighting. To mitigate the interference of confounding (non-causal) factors and spurious correlations, while enhancing model generalization, we introduce causal learning concepts into the CVGL task. Drawing on previous causal modeling work and considering the characteristics of street images, we establish the first Structural Causal Model (SCM) for CVGL, and perform causal intervention guided by the SCM. Since domain-specific (non-causal) signals often reside in extreme low and high frequency bands, while mid-frequency components typically preserve structure-relevant, discriminative information [7] (Figure 2), we design our Causal Feature Extractor as a do-operation in the frequency domain. This allows us to implement a back-door adjustment -akin to causal interventions -that increases the model's attention to causal factors while reducing interference from non-causal factors. To further enhance the model's geometric awareness and mitigate the impact of large viewpoint gaps, we propose the Geometric Topology Fusion (GT Fusion) module, which robustly integrates BEV road topology into street features, leveraging clearer and more localized road topology compared to complex street images.

At the feature level, conventional pooling layers often fail to capture rich semantic cues. To address this, we propose a DA Pooling module that dynamically refines feature representations, enabling the model to capture more context-aware information across diverse scenes and viewpoints.

In summary, our main contributions are:

- We are the first to introduce causal learning concepts into CVGL tasks by applying causal interventions to latent confounding factors, thereby reducing their influence on feature learning. This mechanism enables the model to focus on causally relevant information, such as building structures and road layouts, leading to improved robustness and generalization in complex environments.

- We propose a GT Fusion module that enhances the model's ability to perceive geometric information, mitigating the issue of large viewpoint discrepancies in CVGL tasks and providing more robust localization performance for CVGL.

- We design a DA Pooling to extract rich semantic information and enhance semantic representations across different environments.

Our work highlights the importance of structural reasoning and causal robustness in bridging the cross-view domain gap, setting a new direction for future research in geo-localization.

## 2 Related Work

### 2.1 Cross-view Geo-localization

**Contrastive Learning-based methods.** Contrastive learning has been widely applied in cross-view geolocation tasks [1; 25; 26; 21; 28; 31]. It helps mitigate feature distribution discrepancies between different viewpoints. For example, ConGeo [13] leveraged both single-view and cross-view contrastive losses while incorporating view-specific augmentation strategies. This effectively extracts robust feature representations, enabling the model to maintain high matching accuracy despite viewpoint limitations and orientation deviations. Moreover, Sample4Geo [1] proposed a simplified yet effective contrastive learning framework with a symmetric InfoNCE [19] loss, which fully utilizes all negative samples to accelerate model convergence.

**Incorporation of Geometric Information.** To address the challenges posed by drastic viewpoint variations, some approaches focused on extracting geometric layout information or leveraging BEV images to enforce geometric consistency constraints. For instance, GeoDTR [33] employed a geometric layout extractor to learn spatial correlations between aerial and street features, preventing overfitting to low-level details. Similarly, EP-BEV [28] and HC-Net [21] integrated BEV representations into cross-view geolocation to bridge the substantial differences between views. EP-BEV utilized a dual-branch structure to impose geometric consistency constraints, while HC-Net [21] directly reformulated cross-view geolocation as an image alignment problem.

### 2.2 Causality in Computer Vision

This limitation underscores the motivation for causal inference in visual learning: relying solely on statistical correlations in data is insufficient for reliably predicting counterfactual outcomes and may amplify spurious associations. Causal inference, by modeling the underlying data-generating mechanisms, aims to isolate invariant causal factors and thereby enhance generalization to unseen domains and conditions. To mitigate the interference of non-causal features and extract invariant causal representations, causal mechanisms have been widely adopted in computer vision [18; 3; 17; 30; 6].

In cross-view geo-localization tasks, it is essential to first establish causal relationships and then apply causal inference—including interventional estimation and counterfactual analysis—to eliminate confounding contextual factors and domain shifts. This approach enhances model robustness against domain variations and weather conditions, which pose significant challenges in this task. Drawing on previous causal modeling work [12; 10] and considering the characteristics of street images, we formally define the causal relationships cross-view geo-localization as shown in Figure1, and employ interventional estimation to block the direct influence of confounders, significantly improving generalization. There are two common methods for causal intervention: front-door adjustment and back-door adjustment. Front-door adjustment is used when non-causal factors (confounders) are unobserved, requiring the introduction of an intermediate variable $M$ to reduce the influence of confounding factors. When confounders are observable, back-door adjustment is used, where confounding factors are directly intervened to reduce their impact.

## 3 Method: CLGT

This paper proposes a novel framework for cross-view geo-localization. A multi-head attention-based fusion module, which robustly integrates BEV features into street features via cross-attention and dual dynamic fusion, enforces geometric consistency constraints. To enhance causal features in street representations while mitigating the interference of non-causal features, we employ causal inference-based estimation and intervention. Furthermore, we introduce a DA pooling module to refine the fused features with rich semantic information. The overall model architecture is shown in Figure 3. The following sections provide a detailed introduction to our proposed method.

### 3.1 Preliminary

**BEV Generation.** Various methods exist for generating BEV images, including geometry-based transformations [21; 28], Transformer-based [35], and diffusion-based methods [29]. To balance

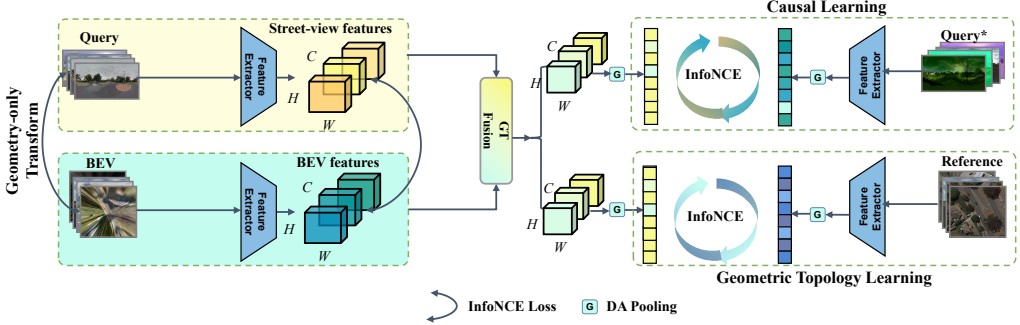

Figure 3: Overview of the proposed Causal Learning and Geometric Topology (CLGT) framework. The road topology information from BEV is fused via the GT Fusion module to obtain the fused features, which are then used for location matching with aerial image features. The causally enhanced street features from the CFE module provide causal supervision, and the DA Pooling module performs final feature extraction.

efficiency and memory use, we adopt the geometric transformation from [21], which directly computes BEV point positions via geometric back-projection from panoramic images. This explicit mapping projects street-view images into BEV space without relying on depth estimation or camera parameters, enabling a simple and efficient street-to-BEV transformation.

**Structural Causal Model.** As illustrated in Figure 1, our SCM is grounded on the following assumptions:

- The street image $X_s$ is mainly generated from two sources: semantic content $C$ and a domain confounder $D$ (e.g., background, lighting), denoted as $C \rightarrow X_s \leftarrow D$.

- The content $C$ contains both discriminative and non-discriminative parts. Together with $D$, the non-discriminative part contributes to the generation of non-causal features $F_n$ via $D \rightarrow F_n \leftarrow C$. The CFE module perturbs a portion of these non-causal components.

- The causal features $F_c$ are derived from the discriminative part of $C$ via $C \rightarrow F_c$.

- The full feature representation $Z = \{F_c, F_n\}$ influences the final prediction $Y$ via $Z \rightarrow Y$, where $Y$ is the matching label.

The overall cross-view matching process can be expressed as $X_a \rightarrow f(X) \rightarrow Y \leftarrow f(X) \leftarrow X_s$, where $X_a$ denotes the aerial image. In this context, the SCM formulation $Z \rightarrow Y$ is a causal abstraction of the model's computational path $X_s \rightarrow f(X) \rightarrow Y$.

## 3.2 Causal Learning

The complete process of Causal Learning is illustrated in the top-right corner of Figure 3, where $Query^*$ is obtained through the Causal Features Extractor.

**Causal Features Extractor.** As shown in Figure 2, roads and buildings in street-view images tend to occupy the mid-frequency spectrum, while style variations are concentrated in the high and low ends, respectively. This aligns with the nature of CVGL, where structural elements are crucial for localization, and view-specific cues often act as noise. To isolate task-relevant features, we leverage the Discrete Cosine Transform (DCT) in our Causal Feature Extractor. Then our Content-aware Mask (CaM) constructs three concentric circular masks with initial radii of $r_1$, $r_2$, and $r_3$, dividing the frequency spectrum into four regions. Unlike prior work [23] that used fixed spectral thresholds, these radii are linearly increased based on image gradient magnitude (via Sobel operator), so that images with stronger gradients preserve more mid-frequency components, enabling better retention of causal information. Larger radii correspond to stronger Gaussian perturbations in outer frequency bands. This allows the model to adaptively preserve mid-frequency, causal components while suppressing non-causal signals. The masked frequencies are then transformed back via inverse DCT. The entire

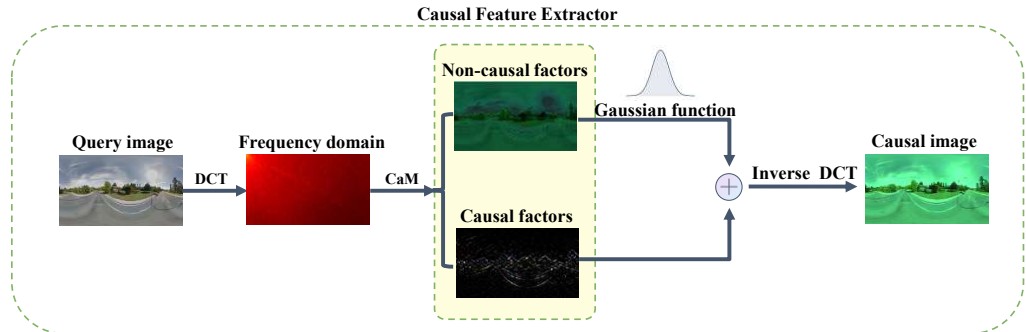

**Causal Feature Extractor**

Figure 4: Illustration of the Causal Feature Extractor (CFE). The input image is transformed into the frequency domain via Discrete Cosine Transform (DCT). A Content-aware Mask (CaM) strategy dynamically separates mid-frequency causal components from low- and high-frequency non-causal components. A Gaussian function is applied to only to the non-causal components (e.g., lighting and brightness) to reduce their influence. Both causal and non-causal parts are then reconstructed via inverse (IDCT) to obtain the causally enhanced image.

process of CFE is shown in Figure 4. The Causal Features Extractor is defined as:

$$CFE(x) = \mathcal{F}' \left( \underbrace{(1 - M(r)) \cdot \mathcal{F}(x)}_{\text{Causal}} + \underbrace{\mathrm{G}(M(r) \cdot \mathcal{F}(x))}_{\text{Non-Causal Randomized}} \right) \tag{1}$$

where $\mathcal{F}$ denotes the Discrete Cosine Transform and $\mathcal{F}^{-1}$ is its inverse. $M(r)$ is a content-aware circular band-pass mask with radius $r$. $G(\cdot)$ denotes a randomization function, defined as $G(X) = X \cdot (1 + \mathcal{N}(0, 1))$.

After obtaining $Query^*$ through the CFE module ($\mathrm{do}(X_s := X_s^*)$), we impose a supervision loss between the causally enhanced features derived from $Query^*$ and the fused features to weaken the path $C \to X_s \to f(X) \to Y$ (where $C$ denotes confounding variables that influence the generation of $X_s$), achieving the effect similar to back-door adjustment in causal interventions.

### 3.3 Geometric Topology Learning

To guide the fusion of street and BEV features, we propose the GT Fusion module. This module effectively leverages the BEV road topology information while enriching the street features output by the backbone, dynamically injecting road topology information without compromising the street details.

**GT Fusion.** As shown in Figure 5 (left), our fusion module first applies a $3 \times 3$ depthwise convolution to backbone outputs $X_b, X_s \in \mathbb{R}^{C \times H \times W}$ to extract local features, maintaining the feature shape. To capture global context, instead of the common Spatial Reduction Attention (SRA) [11], which disrupts boundary spatial structure via non-overlapping token reduction, we adopt Overlapping Spatial Reduction (OSR) to preserve spatial coherence. Finally, $X_s$ acts as the **query** and $X_b$ as **key** and **value** in a cross-attention module, effectively fusing street-view and BEV features.

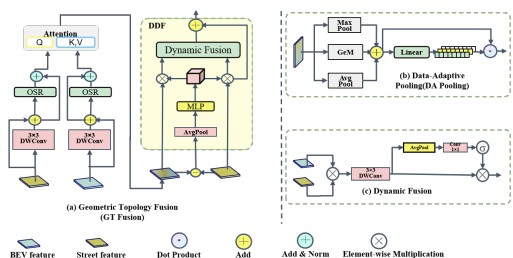

Figure 5: Overview of the GT Fusion and DA Pooling modules. GT Fusion uses cross-attention to exchange semantic information between street and BEV features, then uses Dual Dynamic Fusion (DDF) to enhance fusion robustness. DA Pooling employs a gating mechanism to adaptively weight features, highlighting the most informative ones.

Inspired by [4], we further introduce a Dual Dynamic Fusion (DDF) strategy to robustly integrate the original street features (denoted as $\mathbf{F}^s$) and

their geometry-enhanced counterparts(denoted as $\mathbf{F}^{\mathrm{g}}$). DDF is defined as:

$$
\begin{aligned}
\mathbf{w} &= \sigma\left(\gamma\left(\mathrm{AvgPool}(\mathbf{F}^{\mathrm{s}} + \mathbf{F}^{\mathrm{g}}))\right)\right) \\
\mathrm{Adaptive}(\mathbf{F}) &= \sigma(\mathbf{W} \cdot \mathrm{AvgPool}(\mathbf{F})) \cdot \mathbf{F} \\
\mathbf{F}_{\mathrm{fused}} &= \mathrm{Adaptive}\left(\mathrm{Conv}([\mathbf{w} \cdot \mathbf{F}^{\mathrm{s}},\ (1 - \mathbf{w}) \cdot \mathbf{F}^{\mathrm{g}}])\right)
\end{aligned} \tag{2}
$$

where $\mathbf{F}^{\mathrm{s}}$ denotes the original street-view feature. $\mathbf{W}$ is a $1 \times 1$ convolution, while Conv refers to a $3 \times 3$ convolution. $[\cdot, \cdot]$ denotes the concatenation operation along the channel dimension. $\gamma$ is a linear transformation, and $\sigma$ denotes the sigmoid activation function. The GT Fusion module can be formulated as follows:

$$
X'_s = \mathrm{proj}(X_s) + X_s, X'_b = \mathrm{proj}(X_b) + X_b \tag{3}
$$

$$
Q = \mathrm{LM}(\mathrm{OSR}(X'_s) + X'_s); K, V = \mathrm{LM}(\mathrm{OSR}(X'_b) + X'_b) \tag{4}
$$

$$
Z = \mathrm{Softmax}\left(\frac{QK^{\top}}{\sqrt{d}} + B\right)V, \mathbf{F}_{\mathrm{fused}} = \mathrm{DDF}(X_s, Z) + X_s \tag{5}
$$

where $proj$ refers to the $3 \times 3$ depthwise convolution, with $X_s$ and $X_b$ representing the street and BEV features output by the backbone, respectively. LM denotes a Layer Normalization. $OSR$ stands for Overlapping Spatial Reduction, which is used for global information extraction. $B$ is a relative position bias matrix that encodes the spatial relationships within the attention maps, $d$ represents the number of channels in each attention head, and DDF denotes a dual dynamic fusion module.

**Data-Adaptive Pooling.** As shown on the right side of Figure 5, our Data-Adaptive Pooling module improves upon conventional pooling methods such as global max pooling, global average pooling, and adaptive pooling. These traditional pooling techniques, when used as the final feature aggregation step, fail to effectively capture the rich semantic information of the features. To enhance the representation capability, we combine global max pooling, global average pooling, and geometric mean (Gem) pooling into a single Gate Pooling module. This allows the model to autonomously learn the pooling output, and improve the quality of the final token, thus enhancing both the model's representational power and robustness, which can also be formulated as:

$$
F_{max} = MaxPool(Z), F_{avg} = AvgPool(Z), F_{gem} = Gem(Z) \tag{6}
$$

$$
F_{output} = Gate(Linear(F_{max} + F_{avg} + F_{gem})) \tag{7}
$$

where $MaxPool$ is global max pooling, $AvgPool$ is global average pooling, $Gem$ denotes a geometric mean pooling, Linear is a Linear Functiion, and $Gate$ denotes a gating mechanism.

### 3.4 Loss Function

We apply InfoNCE loss between the fused features and the aerial image features, which serves as the primary supervision signal to optimize our model. The InfoNCE loss is defined as:

$$
\mathcal{L}(f, S)_{\mathrm{InfoNCE}} = -\log \frac{\exp(f \cdot r_+/\tau))}{\sum_{i=0}^{R} \exp(f \cdot r_i/\tau))} \tag{8}
$$

where $f$ denotes the fused feature guided by the query street image, and $S$ is the set of encoded aerial images with one positive $r_+$ matching $f$. The InfoNCE loss computes the dot-product similarity between $f$ and each $r_i$, maximizing similarity with $r_+$ and minimizing it with negatives. The temperature $\tau$ controls distribution sharpness and can be fixed or learned.

As stated in Section 3.2, we apply InfoNCE loss between the causally enhanced features and the fused features to achieve a similar effect to back-door adjustment, encouraging the fused features to focus on causal components. Prior to fusion, to encourage the BEV and street features to lie in a geometrically consistent space, we also apply InfoNCE loss between the two. To preserve their complementarity, we apply a scaling factor to control their learning balance. Thus, we obtain the overall loss of CLGT by computing the weighted sum of them as follows:

$$
\mathcal{L}_{CLGT} = \mathcal{L}(f, S)_{infoNCE} + \gamma\mathcal{L}(f, s^*)_{infoNCE} + \alpha\mathcal{L}(s, b)_{infoNCE} \tag{9}
$$

where $\alpha$ and $\gamma$ are scaling coefficients, $s^*$ denotes the causally enhanced street features, $s$ represents the original street features, and $b$ is the BEV feature.

# 4 Experiment

In our evaluation we conduct experiments on four standard benchmarks, namely CVUSA [22], CVACT [8], VIGOR [36] and CVACT_val-C-ALL, CVACT_test-C-ALL, CVUSA-C-ALL [32]. In the subsequent tables we compare our approach with previous work.

## 4.1 Dataset and Evaluation Protocol

**Dataset.** We evaluate our model on three widely-used cross-view geo-localization benchmarks—CVUSA, CVACT, and VIGOR—as well as their robust variants: CVACT_val-C-ALL, CVACT_test-C-ALL, and CVUSA-C-ALL, which introduce various real-world corruptions to test model robustness under challenging conditions. CVUSA and CVACT each provide 35,532 training and 8,884 testing image pairs with a strict 1-to-1 ground-to-aerial correspondence. In addition, CVACT offers an extra 92,802 GPS-tagged query images for large-scale retrieval evaluation, making it suitable for both standard and large-scale testing scenarios. VIGOR is a more challenging benchmark that spans four metropolitan areas—New York, Seattle, San Francisco, and Chicago—and includes 105,214 query and 90,618 reference images. Unlike CVUSA and CVACT, VIGOR introduces a harder retrieval setup by assigning each query one true positive and three semi-positive samples, thus increasing the difficulty of discriminative matching. It also supports both same-city and cross-city evaluation settings to assess generalization. To further assess model robustness in realistic conditions, we employ corruption-augmented datasets: CVACT_val-C-ALL, CVACT_test-C-ALL, and CVUSA-C-ALL. These variants simulate 16 types of visual degradations, generating approximately 1.5 million corrupted images in total. They provide a rigorous benchmark for evaluating the model's ability to maintain performance under various environmental and sensor-induced perturbations.

**Evaluation Protocol.** We adopt Recall@K as the primary evaluation metric, where $K \in \{1, 5, 10\}$, as well as Recall@1%. A query is considered correctly localized if its corresponding aerial image appears among the top-K retrieved candidates for a given street panorama.

## 4.2 Implementation Details

During the retrieval stage, we adopt ConvNeXt-B as the backbone encoder for both street images and aerial images. Our baseline is EP-BEV. To reduce computational cost and memory usage, we set the image resolution to $384 \times 384$, consistent with EP-BEV. The model is optimized using AdamW with an initial learning rate of $0.5 \times 10^{-3}$. We train the network for 40 epochs with a batch size of 128. The training is conducted on eight 32GB NVIDIA V100 GPUs. For both $\alpha$ and $\gamma$ in Equation 9, we set their values to 0.1 to provide auxiliary supervision

Table 1: Ablation study on causal learning: comparisons of performance on the CVACT_val-C-ALL and CVACT_test-C-ALL datasets.

| Model | CVACT_val-C-ALL | | | CVACT_test-C-ALL | | |
|---|---|---|---|---|---|---|
| | R@1 | R@5 | R@10 | R@1 | R@5 | R@10 |
| Ours | **88.68** | **95.58** | **96.66** | **69.06** | **90.12** | **92.70** |
| Only CL | 88.11 | 94.91 | 96.04 | 67.88 | 89.31 | 91.85 |
| Baseline | 85.94 | 94.52 | 95.93 | 64.62 | 87.75 | 90.78 |

without overwhelming the main optimization objective. When we increase the value of $\gamma$, the model performance improves across various datasets. However, to prevent the value from becoming too large and causing model collapse, which would negatively affect the matching between street and aerial images, we set a default value of 0.1, although this collapse was not observed during training. The optimal value is 0.5, and we will also provide hyperparameter experiments and model performance with $\gamma = 0.5$ in the supplementary materials. We set the initial three radii for the content-aware mask to 0.1, 0.3, and 0.6, respectively. We also observe that performance is stable under small variations in the initial radius. Other training settings follow those used in Sample4Geo.

## 4.3 Comparing with State-of-the-art Models

**Cross-view Image retrieval.** As shown in Table 2, our method achieves the best overall performance across CVUSA, CVACT_val, and CVACT_test datasets. Compared with the strong baseline EP-BEV, our model improves Recall@1 from 97.41% to 98.85% on CVUSA, and from 90.61% to 91.97% on CVACT_val. On the more realistic CVACT_test set, we achieve 73.22% Recall@1, surpassing EP-BEV by 1.81% points. These consistent gains demonstrate the effectiveness of our design. The

integration of BEV-based geometric topology helps capture structured layout cues, while the causal learning strategy improves robustness by suppressing spurious visual signals. Together, they enable more discriminative and generalizable representations for cross-view geo-localization. Results on the VIGOR dataset are provided in the supplementary materials.

Table 2: Comparisons with state-of-the-art models on the CVUSA, CVACT_val and CVACT_test datasets. (†methods that use polar transformation.)

| Model | CVUSA | | | | CVACT_val | | | | CVACT_test | | | |
|---|---|---|---|---|---|---|---|---|---|---|---|---|
| | R@1 | R@5 | R@10 | R@1% | R@1 | R@5 | R@10 | R@1% | R@1 | R@5 | R@10 | R@1% |
| SAFA† [14] | 89.84 | 96.93 | 98.14 | 99.64 | 81.03 | 92.80 | 94.84 | - | - | - | - | - |
| LPN [20] | 85.79 | 95.38 | 96.98 | 99.41 | 79.99 | 90.63 | 92.56 | - | - | - | - | - |
| LPN† [20] | 92.83 | 98.00 | 98.85 | 99.78 | 83.66 | 94.14 | 95.92 | 98.41 | - | - | - | |
| DSM [16] | 91.96 | 97.50 | 98.54 | 99.67 | 82.49 | 92.44 | 93.99 | 97.32 | - | - | - | |
| TransGeo [37] | 94.08 | 98.36 | 99.04 | 99.77 | 84.95 | 94.14 | 95.78 | 98.37 | - | - | - | |
| GeoDTR [33] | 93.76 | 98.47 | 99.22 | 99.85 | 85.43 | 94.81 | 96.11 | 98.26 | 62.96 | 87.35 | 90.70 | 98.61 |
| GeoDTR+ [34] | 95.05 | 98.42 | 98.92 | 99.77 | 87.76 | 95.50 | 96.50 | 98.32 | 67.75 | 90.15 | 92.73 | 98.53 |
| GeoDTR† [33] | 95.43 | 98.86 | 99.34 | 99.86 | 86.21 | 95.44 | 96.72 | 98.77 | 64.52 | 88.59 | 91.96 | 98.74 |
| Sample4G [1] | 98.68 | 99.68 | 99.78 | 99.87 | 90.81 | 96.74 | 97.48 | 98.77 | 71.51 | 92.42 | 94.45 | 98.70 |
| ConGeo [13] | 98.30 | - | - | **99.90** | 90.10 | - | - | 98.20 | 71.70 | 98.30 | - | - |
| EP-BEV [28] | 97.41 | 99.40 | 99.60 | 99.76 | 90.61 | 96.57 | 97.32 | 98.71 | 71.41 | 92.38 | 94.37 | 98.77 |
| Ours | 98.73 | **99.71** | 99.80 | 99.84 | 91.61 | 96.93 | **97.72** | **98.77** | 73.03 | 93.03 | 94.81 | 98.63 |
| Ours ($\gamma = 0.5$) | **98.85** | **99.71** | **99.81** | 99.86 | **91.97** | **96.95** | **97.72** | **98.77** | **73.22** | **93.50** | **95.23** | **98.79** |

**Robustness Evaluation.** As shown in Table 3, our method consistently outperforms baselines across robust datasets, with an average improvement of 5.00%. Notably, it achieves a 6.62% gain on CVUSA-C-ALL, highlighting the model's ability to extract localization-relevant cues such as edge textures of buildings and road structures, while suppressing non-causal noise like lighting and weather conditions. On challenging splits such as CVACT_val-C-ALL, CVACT_test-C-ALL, and CVUSA-C-ALL, our approach demonstrates strong robustness by mitigating the impact of 16 common perturbations and improving retrieval accuracy. Furthermore, in cross-dataset evaluation (trained on CVUSA, tested on CVACT), our method improves performance by 5.50% and 2.84% (Table 6), further validating its generalization and robustness under distribution shifts.

Table 3: Comparisons with state-of-the-art models on the CVUSA-C-ALL, CVACT_val-C-ALL and CVACT_test-C-ALL datasets.

| Model | CVUSA-C-ALL | | | | CVACT_val-C-ALL | | | | CVACT_test-C-ALL | | | |
|---|---|---|---|---|---|---|---|---|---|---|---|---|
| | R@1 | R@5 | R@10 | R@1% | R@1 | R@5 | R@10 | R@1% | R@1 | R@5 | R@10 | R@1% |
| CVM-Net [5] | 6.09 | 16.05 | 23.14 | 52.51 | - | - | - | - | - | - | - | - |
| OriCNN [9] | 9.38 | 22.26 | 30.04 | 58.99 | 15.31 | 28.31 | 35.21 | 58.39 | 3.69 | 8.33 | 11.04 | 43.93 |
| SAFA [14] | 63.68 | 78.08 | 82.82 | 93.91 | 56.72 | 73.60 | 78.59 | 91.32 | 31.18 | 52.06 | 58.60 | 90.41 |
| CVFT [15] | 41.05 | 64.01 | 72.64 | 91.37 | 45.69 | 66.45 | 72.97 | 88.38 | 22.82 | 43.48 | 51.07 | 88.99 |
| DSM [16] | 75.27 | 86.26 | 89.42 | 95.07 | 70.04 | 82.81 | 85.86 | 93.51 | 47.13 | 68.41 | 73.52 | 93.18 |
| L2LTR [25] | 87.93 | 95.45 | 97.01 | 99.01 | 82.13 | 93.34 | 94.93 | 98.10 | 57.20 | 82.59 | 87.23 | 98.09 |
| TransGeo [37] | 82.72 | 91.95 | 94.03 | 97.92 | 74.04 | 86.19 | 89.10 | 94.98 | 52.18 | 74.35 | 78.99 | 95.03 |
| GeoDTR [33] | 84.64 | 93.29 | 95.01 | 98.24 | 77.40 | 88.95 | 91.28 | 95.91 | 52.87 | 78.84 | 83.17 | 95.84 |
| EP-BEV [28] | 86.22 | 94.86 | 96.58 | 99.00 | 85.94 | 94.52 | 95.93 | 98.21 | 64.62 | 87.75 | 90.78 | 98.43 |
| Ours | 92.64 | 97.21 | 98.21 | **99.35** | 88.68 | 95.58 | 96.66 | **98.49** | 69.06 | 90.12 | 92.70 | 98.41 |
| Ours ($\gamma = 0.5$) | **92.84** | **97.61** | **98.41** | **99.35** | **89.49** | **95.84** | **96.92** | **98.49** | **69.71** | **91.05** | **93.30** | **98.85** |

Table 4: Ablation study on DA Pooling: comparison with other pooling methods on CVACT.

| Method | CVACT_val | | | CVACT_test | | |
|---|---|---|---|---|---|---|
| | R@1 | R@5 | R@10 | R@1 | R@5 | R@10 |
| DA pooling | **91.61** | **96.93** | **97.72** | **73.03** | **93.03** | **94.81** |
| Gem | 89.24 | 95.64 | 96.65 | 69.78 | 92.18 | 93.78 |
| Avg | 91.19 | 96.52 | 97.33 | 71.58 | 92.52 | 94.51 |
| Max | 90.55 | 96.51 | 97.25 | 70.78 | 92.20 | 94.23 |

Table 5: Ablation study of the CLGT on CVACT_val.

| Method | CVACT_val | | | |
|---|---|---|---|---|
| | R@1 | R@5 | R@10 | R@1% |
| CLGT | **91.61** | **96.93** | **97.72** | **98.77** |
| w/o GT Fusion | 91.24 | 96.77 | 97.41 | 98.65 |
| w/o CFE | 91.01 | 96.66 | 97.51 | 98.64 |
| Baseline | 90.61 | 96.57 | 97.32 | 98.71 |

**Ablation Study.** We conduct comprehensive ablation studies on CVACT_val, CVACT_val-C-ALL, and CVACT_test-C-ALL to evaluate the individual contributions of each proposed component. As reported in Table 5, employing only the Geometric Topology Fusion (GT Fusion) module

yields a Recall@1 of 91.01% on CVACT_val. This result highlights the importance of geometric consistency learning, which enables the model to better capture road layouts and spatial structures, thereby improving the alignment between ground-level and aerial images.When using only the CFE module, the model achieves a Recall@1 of 91.24%, illustrating its capability to suppress spurious correlations and guide the model toward learning causally relevant and semantically stable features. This enhancement plays a crucial role in resisting interference from latent confounders. As shown in Table 4, our DA Pooling consistently outperforms traditional pooling schemes across all evaluation metrics. Specifically, compared to GeM, DA Pooling improves Recall@1 on CVACT by +2.37%, showing its advantage in dynamically emphasizing informative spatial regions. This confirms the importance of adaptivity in multi-view feature aggregation.

Table 6: Results on different cross-view transfer tasks. Each task is evaluated with representative methods. (†Methods using polar transformation.)

| Task | Method | R@1 | R@5 | R@10 | R@1% |
|---|---|---|---|---|---|
| CVUSA → CVACT_val | L2LTR [25] | 47.55 | 70.58 | - | 91.39 |
| | L2LTR† [25] | 52.58 | 75.81 | - | 93.51 |
| | GeoDTR [33] | 47.79 | 70.52 | - | 92.20 |
| | GeoDTR† [33] | 53.16 | 75.62 | - | 93.80 |
| | Samp4G [1] | 56.62 | 77.79 | 87.02 | 94.69 |
| | EP-BEV [28] | 59.32 | 80.79 | 86.02 | 94.69 |
| | Ours | 60.70 | 81.40 | 86.10 | 95.16 |
| | Ours ($\gamma = 0.5$) | **64.82** | **84.38** | **88.77** | **96.16** |
| CVUSA → CVACT_test | L2LTR [25] | - | - | - | - |
| | L2LTR† [25] | - | - | - | - |
| | GeoDTR [33] | 11.24 | 18.69 | 23.67 | 72.09 |
| | GeoDTR† [33] | 22.09 | 32.22 | 39.59 | 85.53 |
| | Samp4G [1] | 27.78 | 52.08 | 60.33 | 94.88 |
| | EP-BEV [28] | 32.68 | 58.62 | 65.34 | 95.21 |
| | Ours | 33.23 | 59.59 | 67.53 | 95.31 |
| | Ours ($\gamma = 0.5$) | **35.52** | **63.37** | **71.40** | **96.35** |

Furthermore, as shown in Table 1, causal learning alone brings significant improvements on the more challenging and corrupted test sets: Recall@1 increases by 2.23% on CVACT_val-C-ALL and by 3.20% on CVACT_test-C-ALL. In addition to Recall@1, other evaluation metrics such as Recall@5 and Recall@10 also show consistent improvements, closely approaching the performance of the full CLGT model. These results confirm the effectiveness of our causal learning strategy in improving robustness under real-world corruptions and diverse input conditions. Overall, the ablation results validate that both modules—GT Fusion and CFE—contribute meaningfully and complement each other in addressing the challenges of cross-view geo-localization.

**Visualization Analysis.** To qualitatively assess the effectiveness of CLGT, we visualize attention heatmaps generated by the baseline and CLGT models on test images from the CVUSA dataset. As shown in Figure 6, we first visualize the heatmaps on clean images. It can be seen that the baseline model's attention is more scattered, even focusing on the sky and other background noise, while CLGT consistently attends to task-relevant information, especially road structures. Under heavy snow conditions, compared to the baseline, the regions attended by our model remain almost unchanged, whereas the baseline's focus is completely misaligned. This shows that our CLGT consistently attends to semantically meaningful structures, such as road intersections and corner layouts, which are more stable across views. This demonstrates the effectiveness of our design in guiding the model to prioritize task-relevant features and suppress distractions, leading to improved cross-view discriminability. Visualizations for other corruption types can be found in the supplementary materials.

**Complexity Analysis.** As shown in Table 7, we report both GFLOPs and average inference time (in milliseconds per batch of 128 images) on the CVACT_val set. The results demonstrate that our method achieves the best R@1 accuracy with only marginal computational overhead compared to other methods.

Table 7: Comparison of GFLOPs and average inference time per batch (batch size = 128) on CVACT_val. Avg inference time (ms) represents the mean time to process one batch.

| Method | GFLOPs | Avg Inference Time (ms) | CVACT_val R@1 |
|---|---|---|---|
| Sample4Geo | **90.54** | **2367.55** | 90.81 |
| EP-BEV | **90.54** | 2396.53 | 90.61 |
| Ours | 90.56 | 2374.76 | **91.61** |

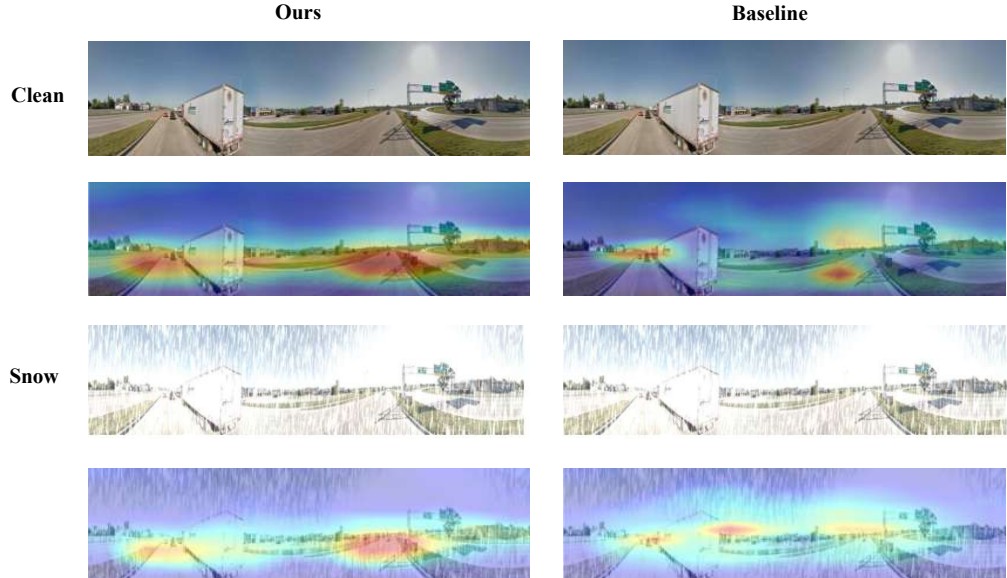

Figure 6: Heatmap visualizations on CVUSA under clean and snow settings. Compared to the baseline, our method focuses more on task-relevant structural information. Under heavy snow conditions, our method remains highly robust, with attention regions largely unchanged, whereas the baseline's attention is completely misaligned.

## 5 Conclusion and Future Work

In this work, we present a novel cross-view geo-localization framework that integrates BEV-street view fusion with causal learning mechanism. Unlike previous methods that utilize BEV merely as an auxiliary representation, our approach enables feature-level interaction that effectively and robustly incorporate road topology. To further improve generalization, we introduce a causal intervention module, thereby enhancing filters out non-causal information and enhances model robustness under various conditions. Experimental results on both standard and challenging datasets demonstrate consistent performance gains. Nonetheless, the BEV representations derived from geometric transformations contains considerable noise, which limits further the performance improvements. Future work will explore more advanced causal inference strategies tailored to the dynamics of complex cross-view localization tasks.

## 6 Acknowledgments

This work was supported in part by Shenzhen Science and Technology Program under Grant JCYJ20240813142510014 and Grant 20220810142553001, in part by the Key Project of Department of Education of Guangdong Province under Grant 2023ZDZX1016, and in part by the National Natural Science Foundation of China under Grant 62072318 and Grant U22A2097.

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
