# OpenReview forum: "CVGL: Causal Learning and Geometric Topology"
_NeurIPS.cc/2025/Conference — NeurIPS 2025 poster_

### Official Review · Reviewer_Pc7t · 2025-06-20

**Clarity:** 3
**Significance:** 3
**Originality:** 3
**Rating:** 4
**Confidence:** 2

**Summary:**

This paper presents a novel framework for cross-view geo-localization. The primary goal is to improve matching accuracy between street-level and aerial images, especially under challenging real-world conditions.  The authors introduce three main components, namely GT Fusion, CFE and DA-Pooling module for this task, and the proposed method is evaluated on several standard and corruption-augmented benchmarks, where it demonstrates state-of-the-art performance, particularly in robustness against visual corruptions.

**Questions:**

1. The paper's central claim is predicated on a causal framework, yet the connection between the SCM (Figure 1) and the CFE is not explicit. Please bridge this gap: How does the do()-operation translate to Equation 1? Why is this a principled causal intervention?

2. Could you provide a comparative analysis of inference time and computational overhead against baselines and other top methods?

3. The CFE relies on the heuristic that factors are separable in the frequency domain. Could you provide a more rigorous analysis or empirical validation for this assumption?

If the authors can convincingly address them, It would be likely that I raise my score.

**Ethical Concerns:**

["NO or VERY MINOR ethics concerns only"]

**Final Justification:**

the rebuttal has successfully addressed my most critical technical and empirical questions. The reasons to accept now outweigh the remaining minor concerns about the framing of the causal claims. I have therefore raised my score.

Initially, I had three main reservations: (1) the tenuous link between the formal causal model (SCM) and the implementation of the Causal Feature Extractor (CFE); (2) the lack of empirical validation for the key assumption that causal and non-causal features are separable in the frequency domain; and (3) the absence of a computational cost analysis.

The authors have successfully resolved concerns (2) and (3) by providing new experimental results.
- The new data provided in Table I of the rebuttal demonstrates that the proposed method achieves its state-of-the-art accuracy with only marginal computational overhead compared to key baselines like EP-BEV. This fully addresses my concern about the method's practical viability.
- More importantly, the new ablation study presented in Table II provides strong empirical validation for the CFE's design. By showing that performance drops when perturbing the mid-frequency bands (which are assumed to be causal), the authors give significant credence to their heuristic. This moves the CFE's design from being seemingly ad-hoc to being an empirically grounded and well-motivated engineering choice.

Regarding the primary concern about the causal framework, the authors' rebuttal clarifies their reasoning significantly. While I still find the link to the formal "back-door adjustment" principle to be more analogical than rigorous, the authors have made it clear that their work is guided by causal *thinking* to motivate a novel data augmentation and feature consistency scheme. The framing could be more cautious, but the underlying idea is novel for this domain and, as shown by the new ablation, empirically sound.

**Limitations:**

Yes

**Paper Formatting Concerns:**

No formatting concerns found.

**Quality:**

2

**Strengths And Weaknesses:**

# Strengths

- **Originality**: The introduction of a causal learning paradigm to CVGL is a significant conceptual contribution, establishing a new research avenue by framing robustness as the mitigation of confounders.

- **Significance & Quality**: The extensive experimental evaluation shows compelling state-of-the-art results, especially on corruption-augmented datasets, providing robust empirical evidence for the proposed components' efficacy.

- **Clarity**: the manuscript is generally well-written.

# Weaknesses

- The paper's central claim of applying causal inference seems undermined by the tenuous link between the formal SCM and the CFE's implementation. The manuscript does not establish how the CFE's frequency-filtering constitutes a principled do()-intervention. Consequently, the method appears to be an effective frequency-based augmentation presented with causal terminology, rather than a rigorous application of causal inference.

- The CFE operates on an simplified heuristic that causal and non-causal features are separable in the frequency domain (lines 139-142) with only qualitative supports. The paper lacks substantive analysis to validate this critical assumption, leaving the CFE's foundation to appear more ad-hoc than theoretically grounded and thereby undermining its claim to causality.

- The manuscript omits analysis of inference time and computational cost, precluding an assessment of the method's practical viability for applications like autonomous navigation.

---

> ### Author Rebuttal · Authors · 2025-07-30
>
> ## Response to Reviewer 4 (R4)
>
> We thank the reviewers for their time and valuable feedback. We appreciate the thoughtful comments and constructive suggestions, which help us improve the clarity and depth of our work.
>
> ### **Q1.** Regarding the SCM and causal intervention details.
>
> **A1:**  We thank the reviewer for raising this important question. Our paper’s causal framework is built on a Structural Causal Model (SCM) that formally characterizes how spurious correlations may arise in cross-view image matching. Below, we clarify the components of the SCM, how it connects to our CFE module, and why our methods corresponds to a principled causal intervention.
>
> ​	**SCM Overview.**
> ​		As illustrated in Figure 1, our SCM is grounded on the following assumptions:
>
> - The street image $X_s$ is mainly generated from two sources: the semantic content $C$ and a domain confounder $D$ (e.g., background, lighting). This is denoted as $C \rightarrow X_s \leftarrow D$.
> - The content $C$ includes both discriminative and non-discriminative information. Together with $D$, the non-discriminative part contributes to the generation of non-causal features $F_n$ via the path $D \rightarrow F_n \leftarrow C$. The proposed **CFE** module attempts to introduce frequency-domain perturbations to part of the non-causal components, which are composed of the non-discriminative part of $C$ and $D$.
> - The causal features $F_c$, on the other hand, are derived from the discriminative part of $C$ through $C \rightarrow F_c$.
> - The full feature representation $Z$ of the street image in the model consists of both causal and non-causal features, which together influence the final prediction $Y$ through $Z \rightarrow Y$, $Y$ is the label indicating whether the two views match.
>
> The overall cross-view matching process is expressed as $X_a \rightarrow f(x) \rightarrow Y \leftarrow f(x) \leftarrow X_s$, where $X_a$ denotes the aerial image.  In this context, the SCM formulation $Z \rightarrow Y$ is a causal abstraction of the model’s computational path $X_s \rightarrow f(X) \rightarrow Y$.
>
> ​	**Causal Intervention via Back-door Adjustment.**
>
> ​	**To avoid confusion, we clarify that in this context, $C$ denotes confounding variables that influence the generation of street-view images $X_s$.**
>
> ​	Our core objective is to reduce spurious correlations by mitigating the influence of non-causal factors through causal intervention. As explained in our SCM, we aim to intervene on the non-causal components of the street image $X_s$, in order to weaken their impact along the spurious path $C \rightarrow X_s \rightarrow f(x) \rightarrow Y$. To achieve this, we adopt a **do-operation** that is concretely implemented by our **CFE module**, formalized in Equation 1.
>
> Specifically, prior work ([7]) has revealed that non-causal factors in natural images tend to exhibit identifiable distributions in the frequency domain. Based on this observation, we design the CFE module to carry out causal intervention directly on the frequency components of $X_s$. The procedure is as follows:
>
>  (1) We first transform the input street image $X_s$ into the frequency domain using DCT;
>
>  (2) A content-aware mask is then applied to separate the frequency components into causal and non-causal parts;
>
>  (3) We perturb only the non-causal frequency components, then add them back to the causal components;
>
>  (4) Finally, we apply inverse DCT to obtain a causally enhanced image $X_s^*$.
>
> Consequently, the perturbation applied to the non-causal components, expressed as $\text{do}(X_s := X_s^*)$, is concretely instantiated in Equation 1, which formalizes the design of our CFE module.
>
> As for **why this constitutes a principled causal intervention**: in SCM, causal intervention requires directly manipulating one or more variables to disrupt the direct path ($C \rightarrow X_s \rightarrow f(X) \rightarrow Y$). In our case, the CFE module applies targeted perturbations to the non-causal components of $X_s$, and further introduces a supervision loss that encourages the fusion feature to align with the causally-enhanced one. This directly intervenes on the path $C \rightarrow X_s \rightarrow f(X) \rightarrow Y$, aligning with the **back-door adjustment** principle in causal inference, and thus qualifies as a principled causal intervention.
>
> **Supplement: Front-door and Back-door Adjustment.**
>
> - **Front-door adjustment** is used when non-causal factors (confounders) are *unobserved*. In such cases, an intermediate variable $M$ is introduced to mediate the causal effect, forming pathways such as $C \rightarrow X \rightarrow M \rightarrow Y$ or $C \rightarrow M \rightarrow X \rightarrow Y$. Intervening on $M$ can help isolate the causal effect of $C$ on $Y$ despite the hidden confounder.
> - **Back-door adjustment** applies when the confounders are *observable or controllable*. In our setting, since we define and perturb the non-causal components  explicitly, we fall into the latter case. We directly perturb the back-door path $C \rightarrow X \rightarrow Y$ by intervening on the non-causal factors—this aligns with the classical back-door criterion.
>
> In both cases, $C$ denotes a non-causal information, and our intervention aims to suppress the influence of spurious pathways on the final prediction $Y$.
>
> We will clarify this explanation further in the final version.
>
> ### **Q2.** Regarding the train and inference time.
>
> **A2:**  We thank the reviewer for the valuable suggestion. Following this recommendation (also raised by Reviewer 2), we conducted a comparative analysis of computational cost and inference speed against representative baselines.
>
> ​	As shown in **Table Ⅰ**, we report both GFLOPs and average inference time (in milliseconds per batch of 128 images) on the **CVACT_val** set. The results demonstrate that our method achieves the best R@1 accuracy with only marginal computational overhead compared to other methods.
>
> **Table Ⅰ**. Comparison of GFLOPs and average inference time per batch (batch size = 128) on CVACT_val. Avg inference time (ms) indicates the mean time to process one batch.
>
> | Method     | GFLOPs    | Avg inference time/ms | CVACT_val R@1 |      |
> | ---------- | --------- | --------------------- | ------------- | ---- |
> | Sample4Geo | **90.54** | **2367.55**           | 90.81         |      |
> | EP-BEV     | **90.54** | 2396.53               | 90.61         |      |
> | Ours       | 90.56     | 2374.76               | **91.61**     |      |
>
> We emphasize that our method maintains comparable inference efficiency to lightweight models (Sample4Geo, EP-BEV), while achieving consistently better accuracy. This confirms the effectiveness of our design in balancing accuracy and computational cost.
>
> ### **Q3.** Regarding the definition of causal and non-causal factors on the frequency spectrum.
>
> **A3:**  We appreciate the reviewer’s thoughtful question. Rather than relying on a heuristic assumption, in Line 40 of our paper, we cite findings that the extremely high and low frequency components in the image often contain the majority of non-causal (domain-specific) information [7]. In contrast, the mid-frequency components typically preserve semantic structures like building shapes and road layouts. In our task, we observe that these structures are precisely the key cues for cross-view matching. Therefore, we follow prior empirical observations to motivate the frequency-domain operation in our CFE module, aiming to preserve such causal information while perturbing the non-causal parts. To make this assumption more transparent, we will include in the final version additional visualizations that highlight the separability of causal and non-causal factors in the frequency spectrum.
>
> ​	We first apply the Discrete Cosine Transform (DCT) to convert the input image into the frequency domain, where ultra-low frequencies lie in the top-left corner and ultra-high frequencies in the bottom-right. To isolate mid-frequency content, we employ a content-aware mask module that constructs several concentric circular masks with radii dynamically adjusted based on the gradient magnitude of each image. After masking, the causal and non-causal frequency components are processed separately: the non-causal components are perturbed with additive noise, while the causal components remain untouched. Both parts are then summed and transformed back to the spatial domain via inverse DCT.
>
> ​	We verify this frequency-domain separability assumption through an ablation study by perturbing the mid-frequency bands instead of the high/low ones. As shown in **Table Ⅱ**, this results in a performance drop, indicating that mid-frequency regions are more likely to encode causal (task-relevant) features. This observation aligns with our core motivation: to enhance causal representation by perturbing frequency regions that predominantly contain non-causal signals.
>
> **Table Ⅱ.** Performance comparison when perturbing mid-frequency components instead of non-causal  regions. The results show a noticeable performance drop, validating that mid-frequency regions encode causal information.
>
> |                               |      | CVACT_val-C-ALL |           |           |           |
> | ----------------------------- | ---- | --------------- | --------- | --------- | --------- |
> | **Frequency Band Perturbed**  |      | R@1             | R@5       | R@10      | R@1%      |
> | low and high frequency (Ours) |      | **88.68**       | **95.58** | **96.66** | **98.49** |
> | Middle-frequency              |      | 87.91           | 94.83     | 95.92     | 98.20     |

---

> > ### Comment · Reviewer_Pc7t · 2025-08-01
> >
> > Thank you for your detailed and thorough rebuttal. It has convincingly addressed my main concerns, and I have increased my score. Although the empirical evidence and the novelty of the approach are strong, I still think the connection to formal causal theory is more analogical than rigorous, and thus could be framed more cautiously.

---

> ### Author Response · Authors · 2025-08-02
> **Appreciation for Constructive Feedback and Score Adjustment**
>
> Dear Reviewer Pc7t,
>
> We sincerely thank the reviewer for the constructive comments and for increasing the score. We greatly appreciate the recognition of the empirical strength and novelty of our approach. We also acknowledge the valuable suggestion regarding the connection to formal causal theory. In the final version, we will revise our presentation to more cautiously and precisely frame this connection. The reviewer’s feedback has been instrumental in helping us improve the quality and clarity of our work. We wish the reviewer continued success and a pleasant academic season.

---

### Official Review · Reviewer_YUub · 2025-06-30

**Clarity:** 2
**Significance:** 2
**Originality:** 3
**Rating:** 3
**Confidence:** 4

**Summary:**

In this paper, the author introduces Causal Feature Extractor, Geometric Topology Fusion via BEV, and DA Pooling for cross-view matching.
My main concern is the design of Structural Causal Model. More illustrations are needed.
Figure 3 is confusing. What is structural causal components and redundant non-causal components in the figure 3 ?
The Causal Feature Extractor is a data-augmentation method, but it is not neccessary better than the conventional data agumentation.
The Geometric Topology Fusion is similar to Qformer, which leverage street feature as query, and the transformed BEV feature as Key and Vale.
The DA pooling is incremental to use three kinds of pooling together.

**Questions:**

Please see the weakness.

**Ethical Concerns:**

["NO or VERY MINOR ethics concerns only"]

**Final Justification:**

Causality part still does not convince me.
The cause is not well-defined.
The association is not intuitive.

I can understand that we need to remove the noise, and do the augmentation. But I do not think it needs  ``causality'' to explain it. So it is somehow over-claimed.

If we only consider the noise and augmentation, the proposed module is not new.

**Limitations:**

Yes.

**Quality:**

2

**Strengths And Weaknesses:**

In this paper, the authors propose a novel framework for cross-view geo-localization that integrates a Causal Feature Extractor (CFE), Geometric Topology Fusion (GT Fusion) using BEV, and a Data-Adaptive Pooling (DA Pooling) module.

Strengths:

1. The paper addresses a timely and challenging problem in cross-view geo-localization and presents a complete pipeline with causal and geometric components.

2. The experimental results are strong across multiple datasets, including corrupted variants, demonstrating robustness.

3. The motivation for introducing causal learning to suppress non-stable features is well-explained.

Main Concerns:

1. Clarity and Justification of Structural Causal Modeling (SCM):

The design of the Structural Causal Model (SCM) is briefly shown in Figure 1 but lacks sufficient justification and explanation. It remains unclear how the variables in the causal graph are defined and how interventions are operationalized in practice.

Figure 3 refers to “causal” and “non-causal” components based on DCT frequency bands. However, the distinction appears heuristic and lacks empirical or theoretical justification. How confident can we be that mid-frequency bands capture "causal" structures while high/low frequencies are "non-causal"? More concrete illustrations or validation would be helpful.

2. Causal Feature Extractor as Frequency Filtering:

The Causal Feature Extractor appears to perform a kind of adaptive frequency-domain filtering with randomized augmentation. While it is framed as causal intervention, the current formulation is arguably closer to frequency-based data augmentation or regularization. The causal interpretation should be further grounded or clarified.

3. Comparison with Standard Augmentations:

While the causal feature extractor is shown to improve robustness, it's not sufficiently compared with more standard or learned data augmentation techniques (e.g., RandAugment, CutMix, or recent frequency-based domain generalization approaches). This weakens the claim that causal modeling offers unique benefits.

4. Similarity to Existing Architectures:

The GT Fusion module resembles prior cross-attention mechanisms such as Q-former, especially the use of street features as queries and BEV features as key/value. The novelty in the geometric fusion design seems marginal unless better distinguished from prior works.

5. DA Pooling Contribution is Incremental:

The DA Pooling module combines standard pooling types (max, average, and geometric mean) with gating. This is a reasonable design choice but seems incremental in isolation. Ablation results show marginal gain; a clearer motivation or design rationale could strengthen its necessity.

6. Reproducibility and Clarity of Presentation:

Several parts of the method, especially loss terms and the frequency masking strategy, are mathematically specified but not intuitively explained. More illustrations and conceptual diagrams would help readers understand the pipeline.

7. The paper title is different from the title in the Openreview system.

8. Cross-view matching. It would be test on drone-view datasets, like University-1652 and Vigor.

---

> ### Author Rebuttal · Authors · 2025-07-31
>
> ## Response to Reviewer 3 (R3)
>
> We thank the reviewers for their time and valuable feedback. We appreciate the thoughtful comments and constructive suggestions, which help us improve the clarity and depth of our work.
>
> ### **Q1.** Regarding the SCM:
>
> **A1:.**  We thank the reviewer for this important question. We provide detailed clarification of our SCM design and empirical validation of frequency-based separation.
>
> **(a) SCM Design:** Our SCM variable design is not ad hoc, but follows established causal modeling principles. Specifically, we draw from *Visual Representation Learning through Causal Intervention for Controllable Image Editing* (CVPR 2025) and *Causal Representation Learning for Domain Generalization* (CVPR  2022), both of which decompose visual data into causal (semantic content) and non-causal (style or domain-specific) factors. Guided by this framework and the characteristics of street image, we construct the SCM in Figure 1.
>
> ​	Our Structural Causal Model (SCM) is constructed based on the following assumptions:
>
> - The street image $X_s$ is mainly generated from two sources: the semantic content $C$ and a domain confounder $D$ (e.g., background, lighting). This is denoted as $C \rightarrow X_s \leftarrow D$.
> - The content $C$ includes both discriminative and non-discriminative information. Together with $D$, the non-discriminative part contributes to the generation of non-causal features $F_n$ via the path $D \rightarrow F_n \leftarrow C$. The proposed **CFE** module attempts to introduce frequency-domain perturbations to part of the non-causal components, which are composed of the non-discriminative part of $C$ and $D$.
> - The causal features $F_c$, on the other hand, are derived from the discriminative part of $C$ through $C \rightarrow F_c$.
> - Full representation $Z$ consists of both components influencing prediction $Y$: $Z \rightarrow Y$
>
> The overall cross-view matching process is expressed as $X_a \rightarrow f(x) \rightarrow Y \leftarrow f(x) \leftarrow X_s$, where $X_a$ denotes the aerial image.  In this context, the SCM formulation $Z \rightarrow Y$ is a causal abstraction of the model’s computational path $X_s \rightarrow f(X) \rightarrow Y$.
>
> **(b) Frequency-Based Separation Justification:** Our frequency separation is grounded in prior work [7] showing that domain-specific (non-causal) signals often reside in extremely low and high frequency bands, while mid-frequency components tend to preserve structure-relevant, discriminative information. To validate this empirically, we conducted ablation studies perturbing different frequency bands. As shown in Table I, perturbing mid-frequency components significantly degrades performance compared to our default setting, confirming that mid-frequencies encode causal signals crucial for localization while extreme frequencies contain non-causal information.
>
> **Table Ⅰ.** Performance comparison when perturbing mid-frequency components instead of non-causal regions.
>
> |                               |      | CVACT_val-C-ALL |           |           |           |
> | ----------------------------- | ---- | --------------- | --------- | --------- | --------- |
> | **Frequency Band Perturbed**  |      | R@1             | R@5       | R@10      | R@1%      |
> | low and high frequency (Ours) |      | **88.68**       | **95.58** | **96.66** | **98.49** |
> | Middle-frequency              |      | 87.91           | 94.83     | 95.92     | 98.20     |
>
> ### **Q2.** Regarding the Connection Between CFE and Causal Intervention:
>
> **A2:** We thank the reviewer for these important questions. We clarify that our CFE is not merely frequency-based or randomized augmentation, but a principled causal intervention module grounded in our SCM. Our SCM identifies the back-door path $C \rightarrow X_s \rightarrow f(x) \rightarrow Y$, where $C$ denotes spurious domain confounders. The CFE leverages the separability of non-causal factors in the frequency domain, introducing controlled perturbations to these components and applying supervision loss to align geometry-enhanced features with perturbed features. This alignment reduces the model's reliance on spurious correlations, following the back-door adjustment strategy in causal inference.
>
> To demonstrate CFE's effectiveness, we compared it against standard data augmentation techniques. As shown in Table-II below, our method consistently outperforms RandAugment and CutMix across CVACT_val-C-ALL, confirming that CFE offers more than regularization effects. Regarding frequency-based domain generalization methods, most recent works (e.g., DFF and PCNorm) do not provide official implementations, making fair comparison challenging within our experimental timeline. We prioritized reproducible baselines to ensure reliable evaluation. This validates that our causally-motivated intervention provides unique benefits beyond conventional augmentation strategies.
>
> **Table Ⅱ.**  Comparison of CFE with standard data augmentation methods on CVACT_val-C-ALL dataset
>
> |             |      | CVACT_val-C-ALL |           |           |           |
> | ----------- | ---- | --------------- | --------- | --------- | --------- |
> |             |      | R@1             | R@5       | R@10      | R@1%      |
> | Ours        |      | **88.68**       | **95.58** | **96.66** | **98.49** |
> | CutMix      |      | 88.05           | 95.21     | 96.42     | 98.36     |
> | RandAugment |      | 87.95           | 95.20     | 96.13     | 98.28     |
>
> ### **Q3.** Regarding the design of GT Fusion.
>
> **A3:** We appreciate the reviewer’s observation. While our GT Fusion adopts a cross-attention structure, it fundamentally differs from models like Q-former, which use learnable queries for semantic alignment. In contrast, we use real street features to attend to BEV features for geometric alignment, tailored to cross-view localization. Beyond the attention itself, our design incorporates (1) BEV-Specific Robustness: We address BEV feature noise from geometric warping through dedicated DDF module, unlike standard cross-attention designs; (2) Task-Driven Design: We employ depthwise convolution and spatial attention (OSR) before fusion, specifically designed for extreme viewpoint alignment in geo-localization while reducing computational overhead;
>
> ### **Q4.** Regarding the ablation study of DA pooling.
>
> **A4:**  We thank the reviewer for this point. Our DA Pooling addresses standard pooling limitations in cross-view tasks: average pooling oversmooths discriminative features, max pooling is noise-sensitive, and GeM pooling applies uniform behavior across all locations despite offering learnable adjustment.
>
> Our DA Pooling uses adaptive gating to dynamically balance max, average, and GeM pooling based on input content. This is crucial for extracting semantic representations from features enhanced by our causal and geometric strategies—inappropriate aggregation can diminish enhancement benefits. As shown in Table IV, DA Pooling consistently outperforms fixed pooling methods, validating its necessity for our enhancement framework.
>
> **Table Ⅳ.** Ablation study of different pooling strategies on the CVACT dataset.
>
> | Method                | CVACT_val |           |           |           | CVACT_test |           |           |           |
> | --------------------- | --------- | --------- | --------- | --------- | ---------- | --------- | --------- | --------- |
> |                       | R@1       | R@5       | R@10      | R@1%      | R@1        | R@5       | R@10      | R@1%      |
> | **DA pooling (Ours)** | **91.61** | **96.93** | **97.72** | **98.77** | **73.03**  | **93.03** | **94.81** | **98.63** |
> | Gem                   | 89.24     | 95.64     | 96.65     | 98.50     | 69.78      | 92.18     | 93.78     | 98.36     |
> | Avg                   | 91.19     | 96.52     | 97.33     | 98.72     | 72.02      | 92.52     | 94.51     | 98.52     |
> | Max                   | 90.55     | 96.51     | 97.25     | 98.70     | 70.78      | 92.20     | 94.23     | 98.44     |
>
> ### **Q5.** Regarding the frequency masking and loss strategy:
>
> **A5:** We thank the reviewer for the suggestion and will include clearer illustrations in the final version. Our CFE  first apply DCT to convert the image into the frequency domain, where low to high frequencies are spatially distributed (low frequencies lie in the top-left and high frequencies in the bottom-right.). Then, our content-aware mask constructs three concentric circular masks with initial radii of 0.1, 0.3, and 0.6, dividing the frequency spectrum into four regions. These radii are linearly increased based on image gradient magnitude (via Sobel operator), so that images with stronger gradients preserve more mid-frequency components, enabling better retention of causal information. Larger radii correspond to stronger Gaussian perturbations in outer frequency bands.  For losses, we add two auxiliary losses: (1) contrastive loss between street and BEV features for geometric pre-alignment, and (2) contrastive loss between causally enhanced and fused features for causal supervision. Both use small weights to guide training.
>
> ### **Q6.** Regarding additional datasets:
>
> **A6:** We thank the reviewer for this comment. We have provided VIGOR evaluation results in Table 6 of the appendix (omitted here due to space limits). For University-1652, since this dataset uses drone-view inputs and our method relies on BEV features generated via geometric transformation from ground-level street images, we cannot directly apply our framework to drone-view inputs. On VIGOR, our method ranks second among existing approaches. The performance gap is attributed to VIGOR's unique challenges: the one-to-many retrieval protocol and street images with high density of dynamic objects that create noisy BEV representations, degrading geometric cue effectiveness. We plan to explore VIGOR-specific adaptations in future work.

---

> ### Author Response · Authors · 2025-08-04
> **Follow-up on Rebuttal Response and Clarification Inquiry**
>
> Dear Reviewer YUub,
>
> We sincerely thank the reviewer for taking the time to read our rebuttal and acknowledge our response. We hope that our detailed clarifications have adequately addressed the reviewer’s concerns. If there are any remaining questions or points requiring further explanation, we would be happy to provide additional information. We genuinely appreciate the reviewer’s efforts, and we would be grateful if the clarification could be reflected in the final evaluation.

---

> ### Comment · Area_Chair_CWJ1 · 2025-08-05
> **Discussion**
>
> Dear Reviewer YUub,
>
> Could you kindly clarify which of your comments you feel have not been adequately addressed? Additionally, could you specify the areas where the paper still requires significant revision?
>
> I encourage both the reviewers and the authors to actively engage in the discussion to help move the process forward constructively.
>
> Best regards,
> AC

---

> > ### Comment · Area_Chair_CWJ1 · 2025-08-08
> > **Discussion**
> >
> > Dear Reviewer YUub,
> >
> > Could you kindly clarify which of your comments you feel have not been adequately addressed? Additionally, could you specify the areas where the paper still requires significant revision?
> >
> > I encourage both the reviewers and the authors to actively engage in the discussion to help move the process forward constructively.
> >
> > Best regards, AC

---

> > > ### Comment · Reviewer_YUub · 2025-08-09
> > >
> > > Causality part still does not convince me. The cause is not well-defined. The association is not intuitive.
> > >
> > > I can understand that we need to remove the noise, and do the augmentation. But I do not think it needs ``causality'' to explain it. So it is somehow over-claimed.
> > >
> > > If we only consider the noise and augmentation, the proposed module is not new.

---

> > > > ### Author Response · Authors · 2025-08-09
> > > >
> > > > Dear Reviewer YUub,
> > > >
> > > > We clarify that our method is not merely performing noise removal or random data augmentation. Instead, starting from the Structural Causal Model (SCM) perspective, we explicitly leverage the distribution of non-causal factors in the frequency domain and design a content-aware mask to selectively perturb these non-causal components. Following this perturbation, we adopt the backdoor adjustment strategy from causal intervention theory to directly influence the matching pathway from street-view images to satellite images.
> > > >
> > > > That said, we respect your perspective, and we will be more cautious in our wording regarding the definition of “causality” in the final version to avoid any possible overstatement.

---

> > > > > ### Comment · Reviewer_YUub · 2025-08-09
> > > > >
> > > > > Thank you.
> > > > >
> > > > > The paper is quite techincal with empricial findings.
> > > > >
> > > > > My main concern is still that the method and the motivation are **not well-aligned** with your Structural Causal Model (SCM). There are still some gaps.

---

> ### Author Response · Authors · 2025-08-09
>
> Dear Reviewer YUub,
>
> Thank you for your continued engagement and valuable feedback. We appreciate your partial recognition of our method. We clarify that in the Introduction of our paper, we describe that besides the inherent viewpoint challenges, the CVGL task is also affected by various confounding factors. Motivated by this, we designed a Structural Causal Model (SCM) specific to CVGL and leveraged the partial definition of confounders from reference [7]. Based on this SCM, we adopt a causal intervention strategy to reduce the influence of confounding factors. Specifically, we perturb the confounders using the CFE module, and then apply a supervised loss between the causally enhanced features and the fused features to implement the backdoor adjustment, thereby effectively mitigating the confounders’ impact. We appreciate your feedback and will refine our presentation to ensure greater clarity.

---

### Official Review · Reviewer_gb9n · 2025-07-01

**Clarity:** 3
**Significance:** 3
**Originality:** 1
**Rating:** 4
**Confidence:** 4

**Summary:**

The paper tackles cross-view geo-localization which consists in matching street view images to aerial images. This is a challenging task due to the large viewpoint changes between street and aerial images, significant illumination changes as well as potential occlusions. To deal with condition variations and the dynamic nature of the scenes, the authors introduce a causal feature extractor that isolates features relevant for localization such as structural elements while removing distracting features. The causal feature extractor predicts a content aware frequency mask to separate relevant/non-relevant features in the frequency space. Furthermore, street view images are lifted to bird eye view and the respective features from both perspectives are merged through the geometric topology fusion module which creates more robust and discriminative feature maps through. Subsequently, global descriptors are pooled from the fused feature maps through a data-adaptative pooling layer which focuses on semantically rich regions. The resulting localization pipeline is evaluated on two cross-view datasets and their variants which include.

**Questions:**

Overall the motivation behind introducing the street/BEV fusion and the causal mechanism are sound and the resulting method is effective and brings small accuracy gains. However given the high accuracy of the baseline method (EP-BEV) this paper is based upon and the limited novelty, some more explanations and ablations would be required to better grasp the significance of the contributions. (cf weaknesses 2 and 3).

l144.  “our method dynamically adapts to each input by generating a content-aware frequency mask”, does this refer to the radius of the band-pass mask, how is r predicted?

What is the benefit of aligning the street and BEV features through the contrastive loss since the street/BEV features are fused anyway in the subsequent GT fusion module? Also this loss would be applied on the pixel level for dense features, do you randomly sample negative features in the image space?

Could you provide a run time comparison with other baselines both for training and inference?

Cf. weakness 3)

**Ethical Concerns:**

["NO or VERY MINOR ethics concerns only"]

**Final Justification:**

Overall I find the justification/formulation around "geometric topology" and "causality" to be overly pompous at time and make it hard to identify where the contributions come from. For example, the CFE module was introduced in [21] (the overall causal framework was also established in prior works and ends up being a type of image augmentation) and the authors contribution lies in modifying the frequency threshold. Similarly, the representation learning framework follows EP-Bev and the authors introduce the GTFusion module to merge street and BEV features before the contrastive learning objectives.
However these incremental contributions are well motivated and, in the rebuttal, the authors provided additional ablations to support the effectiveness of the GTFusion / CFE and data pooling modules.
The resulting pipeline ends up achieving SOTA accuracy with similar computational efficiency as prior baselines. As such, I believe the reasons to accept the papers outweigh the reasons to reject and I have upgraded my score to "borderline accept".

**Limitations:**

Yes

**Quality:**

2

**Strengths And Weaknesses:**

### Strengths
1) The causal learning mechanism is an effective way of injecting causal information in the feature representation. In addition, the causal feature extraction is an elegant way of dissociating semantically important features from distractors without relying on third party pretrained models for detection or condition correction. Its effectiveness is supported by ablation studies.
2) Augmenting street features with BEV features gives a complementary perspective and makes the resulting representation more robust wrt viewpoint changes and more discriminative which is particularly relevant in the context of cross-view geo-localization.
3) The proposed representation shows consistent accuracy improvements over prior cross-view geo-localization baselines across all datasets.

### Weaknesses
1) The causal feature extraction comes from [21] which limits the contributions of the proposed method. The authors state that the difference lies in their content aware frequency mask as opposed to the fixed thresholds used in [21]. However no explanation is provided as to how the content aware mask is predicted. As it is an important part of the paper, this ought to be explained and the benefits of the content aware mask compared to a fixed bandwidth should be shown.

2) I am skeptical about the benefits of the proposed data-adaptive pooling layer. The geometric mean pooling is already a generalization of global average and max pooling layers, where the learnable parameter balances between max and average pooling behaviors (as such GeM is already “data-adaptive”). I do not see any ablation study to justify the design choice of the proposed data-adaptive pooling layer.

3) While I understand the motivation behind fusing street/BEV features I find the following sentences confusing and it is hard to tie them to the GT fusion module’s architecture.
l.113 “A multi-head attention-based fusion module integrates BEV features into street features, enforcing geometric consistency constraints”, how can multi-head attention enforce geometric constraints?
l.151 “effectively leverages the BEV geometric consistency constraints”, what does the geometric consistency constraints refer to here?
l.152 “dynamically injecting road geometric topology information”, is geometric topology the right term to describe scene layout, doesn’t street view image then also contain geometric topology information?
Description of Fig. 4 “ DFF to inject BEV’s geometric topology”, isn’t the BEV features already integrated through the MHA layers
The GT fusion module should be more ablated and tied to the textual explanations. Ablations could include, w/o DDF, w/o OCR, maybe basic fusion mechanisms (MHA, concatenation, summation).

4) Missing reference for InfoNCE: Oord, Aaron van den, Yazhe Li, and Oriol Vinyals. "Representation learning with contrastive predictive coding."

---

> ### Author Rebuttal · Authors · 2025-07-30
>
> ## Response to Reviewer 2 (R2)
>
> We thank the reviewers for their time and valuable feedback. We appreciate the thoughtful comments and constructive suggestions, which help us improve the clarity and depth of our work.
>
> ### **Q1.** Regarding the GT Fusion and DA pooling.
>
> **1.DA pooling:**
>
> **A1:** We thank the reviewer for pointing this out. While GeM pooling generalizes average and max pooling via a learnable exponent $p$, its adjustment is global and uniform, which limits its ability to capture view-dependent spatial variations.  In contrast, our DA Pooling introduces a channel-specific gating mechanism to adaptively combine avg, max, and GeM, allowing finer spatial sensitivity. As shown in Table I, it consistently outperforms all baselines on CVACT, validating its effectiveness. We will include these results in the final version.
>
> **Table Ⅰ.** Ablation study of different pooling strategies on the CVACT dataset.
>
> | Method                | CVACT_val |           |           |           | CVACT_test |           |           |           |
> | --------------------- | --------- | --------- | --------- | --------- | ---------- | --------- | --------- | --------- |
> |                       | R@1       | R@5       | R@10      | R@1%      | R@1        | R@5       | R@10      | R@1%      |
> | **DA pooling (Ours)** | **91.61** | **96.93** | **97.72** | **98.77** | **73.03**  | **93.03** | **94.81** | **98.63** |
> | Gem                   | 89.24     | 95.64     | 96.65     | 98.50     | 69.78      | 92.18     | 93.78     | 98.36     |
> | Avg                   | 91.19     | 96.52     | 97.33     | 98.72     | 72.02      | 92.52     | 94.51     | 98.52     |
> | Max                   | 90.55     | 96.51     | 97.25     | 98.70     | 70.78      | 92.20     | 94.23     | 98.44     |
>
> **2.Clarifying GT Fusion:**
>
> ​	**A2:**  We thank the reviewer for these insightful comments. Our GT Fusion uses a street-as-query strategy where street features query BEV features via multi-head attention to selectively aggregate relevant geometric information. This design aligns the fusion with the street perspective while leveraging BEV's cleaner geometric representation of scene layout. We acknowledge the terminology inconsistency—"geometric consistency constraints" and "geometric topology" both refer to structural layout cues, and we will unify this language. While street images do contain geometric information, their complexity (dynamic objects, lighting variations) makes it harder for models to focus on structural elements. BEV provides a cleaner, top-down view of these structures, which motivates our fusion approach. Regarding DDF, we clarify it serves for robust fusion by suppressing noisy BEV signals rather than "injecting" geometry, and we will revise the figure caption to reflect this.
>
> **3.Ablation Study of GT Fusion and Its Difference from EP-BEV:**
>
> ​	**A3:** We thank the reviewer for the suggestion to provide more detailed ablations on the GT Fusion module. Following this, we conducted experiments removing key components: without DDF, which normally enhances fusion robustness by reducing the impact of geometric distortions and noise in BEV features, performance noticeably drops under challenging conditions due to increased sensitivity to unreliable BEV information; without OSR, which supplies global context and supports long-range dependencies, the fusion’s ability to capture meaningful correspondences between street and BEV features is weakened.
>
> As shown in Table Ⅱ, the removal of either module results in a consistent degradation across all evaluation metrics, highlighting their complementary roles in ensuring effective and robust fusion.
>
> **Table Ⅱ.**  Ablation Study of GT Fusion by Removing DDF or OSR.
>
> | Method          | CVACT_val |           |           |           | CVACT_test |           |           |           |
> | --------------- | --------- | --------- | --------- | --------- | ---------- | --------- | --------- | --------- |
> |                 | R@1       | R@5       | R@10      | R@1%      | R@1        | R@5       | R@10      | R@1%      |
> | GT Fusion(Ours) | **91.61** | **96.93** | **97.72** | **98.77** | **73.03**  | **93.03** | **94.81** | **98.63** |
> | w/o DDF         | 89.38     | 95.51     | 96.60     | 98.47     | 66.95      | 90.18     | 92.76     | 98.44     |
> | w/o OSR         | 91.22     | 96.52     | 97.45     | 98.59     | 72.61      | 92.60     | 94.62     | 98.51     |
>
> **A4.** **Unlike EP-BEV**, which directly uses potentially noisy BEV features, our method shows stronger robustness under challenging conditions by dynamically integrating BEV geometry into street representations. As shown in Table Ⅲ, our GT Fusion module significantly outperforms EP-BEV on the corrupted CVACT_val-C-ALL set, highlighting its effectiveness in suppressing unreliable BEV cues and reinforcing the novelty of our approach.
>
> **Table Ⅲ.** Robustness Comparison on the Corrupted CVACT_val-C-ALL Dataset
>
> |                |      | CVACT_val-C-ALL |           |           |           |
> | -------------- | ---- | --------------- | --------- | --------- | --------- |
> |                |      | R@1             | R@5       | R@10      | R@1%      |
> | Ours           |      | **88.68**       | **95.58** | **96.66** | **98.49** |
> | Only GT Fusion |      | 87.88           | 94.81     | 96.02     | 98.39     |
> | EP-BEV         |      | 85.94           | 94.52     | 95.93     | 98.21     |
>
> ### **Q2.** Regarding the content-aware mask.
>
> **A5:**  We thank the reviewer for highlighting the need to clarify our content-aware frequency mask. Rather than using a fixed band-pass mask radius, we first apply DCT to convert the image into the frequency domain, where low to high frequencies are spatially distributed (low frequencies lie in the top-left and high frequencies in the bottom-right.). Then, our content-aware mask constructs three concentric circular masks with initial radii of 0.1, 0.3, and 0.6, dividing the frequency spectrum into four regions. These radii are linearly increased based on image gradient magnitude (via Sobel operator), so that images with stronger gradients preserve more mid-frequency components, enabling better retention of causal information. Larger radii correspond to stronger Gaussian perturbations in outer frequency bands. This allows the model to adaptively preserve mid-frequency, causal components while suppressing non-causal signals. The masked frequencies are then transformed back via inverse DCT. We also observe that performance is stable under small variations in the initial radius.
>
> We compared our adaptive frequency mask with a fixed-threshold strategy and found that dynamically adjusting mask radii based on image gradients better preserves causal features. As shown in Table IV, our method consistently outperforms the fixed variant on CVACT_val-C-ALL, demonstrating the importance of adaptivity for robust performance in complex scenes.
>
> **Table Ⅳ.** Comparison between our content-aware frequency mask and fixed-threshold frequency mask on CVACT_val-C-ALL.
>
> |                  |      | CVACT_val-C-ALL |           |           |           |
> | ---------------- | ---- | --------------- | --------- | --------- | --------- |
> |                  |      | R@1             | R@5       | R@10      | R@1%      |
> | Ours             |      | **88.68**       | **95.58** | **96.66** | **98.49** |
> | fixed thresholds |      | 88.28           | 94.92     | 96.11     | 98.41     |
>
> ### **Q3**. Regarding the loss between street and BEV features.
>
> **A6:**  We thank the reviewer for the insightful question. While GT Fusion enables deep interaction between street and BEV features, the contrastive loss serves as a complementary pre-alignment regularizer, encouraging both features to lie in a geometrically consistent space before fusion. To be clear, the contrastive loss does not serve as the main objective, but rather acts as a regularization signal to encourage consistency between the two views during encoding. In practice, we scale the contrastive loss by a small factor (0.1) to ensure it has minimal interference with the fusion learning. We also clarify that negative samples are randomly drawn from feature maps to reduce overhead. As further support, we conducted a sensitivity study on $\alpha$ (see Table Ⅴ). We find that small values (0.1) provide slight regularization benefit, while larger values may interfere with the fusion process, consistent with its intended role as a pre-fusion alignment signal.
>
> **Table Ⅴ. CVACT_val | Effect of Varying α (street & BEV alignment weight)**
>
> |               | R@1       | R@5       | R@10      | R@1%      |
> | ------------- | --------- | --------- | --------- | --------- |
> | 0 (no loss)   | 91.55     | **97.09** | 97.69     | 98.71     |
> | 0.1 (default) | **91.61** | 96.93     | **97.72** | **98.77** |
> | 0.3           | 91.45     | 96.81     | 97.52     | 98.66     |
>
> ### **Q4**. Regarding the train and inference time.
>
> **A7:**  We thank the reviewer for the valuable suggestion. Following it, we conducted runtime comparisons including GFLOPs and average inference time per batch (batch size = 128). The results are shown below:
>
> **Table Ⅵ**. Comparison of GFLOPs and average inference time per batch (batch size = 128) on CVACT_val. Avg inference time (ms) indicates the mean time to process one batch.
>
> | Method     | GFLOPs    | Avg inference time/ms | CVACT_val R@1 |      |
> | ---------- | --------- | --------------------- | ------------- | ---- |
> | Sample4Geo | **90.54** | **2367.55**           | 90.81         |      |
> | EP-BEV     | **90.54** | 2396.53               | 90.61         |      |
> | Ours       | 90.56     | 2374.76               | **91.61**     |      |
>
> We emphasize that our method maintains comparable inference efficiency to lightweight models (Sample4Geo, EP-BEV), while achieving consistently better accuracy. This confirms the effectiveness of our design in balancing accuracy and computational cost.

---

> > ### Comment · Reviewer_gb9n · 2025-08-05
> >
> > Thank you for the comprehensive and thorough rebuttal. Most of my points have been addressed and I have no additional questions at this point.

---

> > > ### Comment · Area_Chair_CWJ1 · 2025-08-05
> > > **Discussion**
> > >
> > > Dear Reviewer gb9n,
> > >
> > > As you mentioned that "Most of my points have been addressed and I have no additional questions at this point", what would be your final rating of the paper?
> > >
> > > I encourage both the reviewers and the authors to actively engage in the discussion to help move the process forward constructively.
> > >
> > > Best regards,

---

> > > ### Author Response · Authors · 2025-08-08
> > > **Follow-up on Discussion and Reviewer Feedback**
> > >
> > > Dear Reviewer gb9n,
> > >
> > > We sincerely appreciate your earlier response stating that most of your concerns have been addressed and that you have no additional questions at this point. We are truly encouraged by your engagement during the discussion phase and your constructive feedback throughout the review process.
> > >
> > > As the discussion period draws to a close, we would be grateful to learn your final thoughts and evaluation of our work. We genuinely hope that our efforts to address your concerns — particularly regarding the novelty demonstrated in our rebuttal — have conveyed the contributions and significance of our study.
> > >
> > > We highly value your perspective, and any further comments or acknowledgment would be deeply appreciated. We also look forward to the possibility of continued discussions beyond the review process.
> > >
> > > Thank you once again for your time and consideration.
> > >
> > > Best regards,

---

### Official Review · Reviewer_W79E · 2025-07-03

**Clarity:** 4
**Significance:** 3
**Originality:** 3
**Rating:** 4
**Confidence:** 5

**Summary:**

This paper aims to address the cross-view geo-localization task where a street-view query image is matched with an extensive database of aerial photos. The drastic viewpoint change, weather change, misalignment, and occlusion between the ground and aerial views make the task challenging. To address these challenges, the authors in this work propose a Causal Learning and Geometric Topology (CLGT) framework, which consists of two novel architectural contributions - a causal feature extractor to focus on stable and task-relevant features, and GT fusion to efficiently inject the Bird’s Eye View (BEV) feature with road topology. The paper also proposes data-adaptive pooling, which outperforms conventional global max pooling, global average pooling, and adaptive pooling on the task at hand. State-of-the-art results on four popular benchmarks prove the efficacy of the proposed CLGT framework.

**Questions:**

(1) The ablation studies in this paper can be improved. I am interested in seeing a systematic comparison of (a) Causal Features Extractor vs Features using an off-the-shelf pre-trained encoder, (b) GT fusion vs normal cross-attention based fusion, and (c) data-adaptive pooling vs traditional global average/max pooling. I would like to know the author's opinion on which model component plays the most crucial role in improving performance. The authors have reported ablative experiments in Table 5, but it is not clear to me what the model exactly looks like for the rows 'w/o GT Fusion' and 'w/o CFE', i.e., when GT fusion and CFE are not used. How are the features extracted and fused in these cases?

(2) The paper lacks sufficient implementation and hyperparameter details. Which hyperparameters contribute most to the training stability?

(3) Another central area of improvement is the qualitative results. The authors should explicitly demonstrate examples where the proposed architectural components outperform previous methods. Since the primary contribution of this work is the novelty of the model, we should examine where the proposed system succeeds but previous systems fail.

(4) I acknowledge the paper achieves SOTA results, but the authors have missed multiple important baselines (including some already published in recognized venues). I request that the authors conduct an extensive literature review and report all peer-reviewed and published baselines that have already been established.

(a) Cross-View Image Sequence Geo-localization. WACV 2023.
(b) GeoDTR+: Toward Generic Cross-View Geolocalization via Geometric Disentanglement. TAPMI 2024.
(c) Unleashing Unlabeled Data: A Paradigm for Cross-View Geo-Localization. CVPR 2024.
(d) Robust Cross-View Geo-Localization via Content-Viewpoint Disentanglement. arXiv 2025.
(e) Where am I? Cross-View Geo-localization with Natural Language Descriptions. arXiv 2025.


Due to these limitations, I recommend a 'Borderline rejection' at this stage, and I will wait for the authors to address the points (1), (2), and (3). I understand that the rebuttal timeframe is limited, but having a clearer picture of which model components are most effective is essential. I will be willing to increase the score if the issues are well-addressed.

**Ethical Concerns:**

["NO or VERY MINOR ethics concerns only"]

**Final Justification:**

As mentioned in the comments, the following concerns are addressed:

(1) Systematic comparison of (a) Causal Features Extractor vs Features using an off-the-shelf pre-trained encoder, (b) GT fusion vs normal cross-attention based fusion, and (c) data-adaptive pooling vs traditional global average/max pooling;
(2) Important Hyper-parameters, and
(3) Adding recent baselines.

There are two important issues the paper still misses out: (1) A non-cherry-picked qualitative comparison with baselines, showing a broad variety of samples where the model performs better than existing models, (2) a honest error analysis, describing where the proposed system fail, and some discussion on future works.

Since both issues can be included in the final manuscript (or, can be analyzed by future researchers if the source code and pre-trained checkpoints are released), I decide to raise my score to a 'borderline accept'.

**Limitations:**

The paper does not include any discussion about the failure modes of the proposed system. The authors should discuss the limitations of the CLGT system in a positive light and indicate how it can be further improved.

**Quality:**

3

**Strengths And Weaknesses:**

(1) The paper is well-written and easy to digest. The authors discuss the challenges of the cross-view geo-localization task and systematically propose the model components to solve them. The architectural details are well-documented.
(2) The strongest point of this paper is the effective results - there are several previous works on cross-view geo-localization reporting results on the CVUSA and CVACT datasets. This paper clearly achieves a state-of-the-art score on the majority of the metrics.

Please find the detailed weaknesses in the 'questions' section.

---

> ### Author Rebuttal · Authors · 2025-07-30
>
> ## **Response to Reviewer 1 (R1)**
>
> We thank the reviewers for their time and valuable feedback. We appreciate the thoughtful comments and constructive suggestions, which help us improve the clarity and depth of our work.
>
> ### **Q1.**  Regarding the ablation studies.
>
> - (a) Causal Features Extractor vs Features using an off-the-shelf pre-trained encoder
>
>   **A1:** We clarify that CFE is not a replacement for the pre-trained encoder, but a lightweight module applied in the frequency domain to suppress non-causal signals. It enhances feature discriminability and generalization by emphasizing causal, structurally relevant components. The transformed image is then fed into a standard encoder (ConvNeXt-B), ensuring our models remain based on off-the-shelf backbones.
>
> - (b) GT fusion vs normal cross-attention based fusion
>
>   **A2:**  We thank the reviewer for the valuable suggestion regarding a more comprehensive ablation of fusion strategies. Following this, we compared our GT Fusion module with two recent cross-attention-based designs fusion: **GeminiFusion**: Efficient Pixel-wise Multimodal Fusion for Vision Transformer [arXiv 2024] and **CrossFuse**: A Novel Cross Attention Mechanism based Infrared and Visible Image Fusion Approach [arXiv 2024]. For GeminiFusion, which outputs dual-view features, we used the street-as-query branch for supervision to ensure fairness.
>
>   As shown in Table I, our method consistently outperforms both alternatives across CVACT, highlighting the effectiveness of our design in integrating BEV geometry with street-view features.
>
>   **Table Ⅰ**: Performance comparison of GT Fusion, GeminiFusion, and CrossFuse on the CVACT dataset.
>
>   | Method          | CVACT_val |           |           |           | CVACT_test |           |           |           |
>   | --------------- | :-------: | :-------: | :-------: | :-------: | :--------: | :-------: | :-------: | :-------: |
>   |                 |    R@1    |    R@5    |   R@10    |   R@1%    |    R@1     |    R@5    |   R@10    |   R@1%    |
>   | GT Fusion(Ours) | **91.61** | **96.93** | **97.72** | **98.77** | **73.03**  | **93.03** | **94.81** | **98.63** |
>   | GeminiFusion    |   89.01   |   95.51   |   96.33   |   98.26   |   69.51    |   92.07   |   93.55   |   98.13   |
>   | CrossFuse       |   90.32   |   96.32   |   97.15   |   98.66   |   70.62    |   92.10   |   94.16   |   98.44   |
>
> - (c) Data-adaptive pooling vs traditional global average/max pooling.
>
>   **A3:** We sincerely apologize for omitting the ablation study on pooling strategies in the main paper, and we greatly appreciate the reviewer for pointing this out. As suggested, we conducted a comparative ablation among GeM pooling, global average pooling (Avg), and global max pooling (Max). Furthermore, we applied our DA Pooling on the EP-BEV branch to demonstrate its effectiveness under varying spatial cues.
>
>   As shown in **Table Ⅱ**, our DA Pooling consistently outperforms traditional pooling schemes across all evaluation metrics. Specifically, compared to GeM, DA Pooling improves Recall@1 on CVACT by +2.27%, showing its advantage in dynamically emphasizing informative spatial regions. This confirms the importance of adaptivity in multi-view feature aggregation.
>
>   We will add these results to the final version .
>
>   **Table Ⅱ.** Ablation study of different pooling strategies on the CVACT dataset.
>
>   | Method                | CVACT_val |           |           |           | CVACT_test |           |           |           |
>   | --------------------- | --------- | --------- | --------- | --------- | ---------- | --------- | --------- | --------- |
>   |                       | R@1       | R@5       | R@10      | R@1%      | R@1        | R@5       | R@10      | R@1%      |
>   | **DA pooling (Ours)** | **91.61** | **96.93** | **97.72** | **98.77** | **73.03**  | **93.03** | **94.81** | **98.63** |
>   | Gem                   | 89.24     | 95.64     | 96.65     | 98.50     | 69.78      | 92.18     | 93.78     | 98.36     |
>   | Avg                   | 91.19     | 96.52     | 97.33     | 98.72     | 72.02      | 92.52     | 94.51     | 98.52     |
>   | Max                   | 90.55     | 96.51     | 97.25     | 98.70     | 70.78      | 92.20     | 94.23     | 98.44     |
>
> - (d) Clarification on model components and ablation settings.
>
>   **A4:** We thank the reviewer for this insightful question. As suggested, we clarify the individual contributions of each module and the ablation settings reported in Table 5. Our framework comprises three core components—GT Fusion, the CFE, and DA Pooling—each designed to address a specific challenge in cross-view geo-localization. Among them, CFE proves most critical to performance, as it directly mitigates the influence of confounding factors and enhances generalization. This is consistently evidenced by the improvements shown in both Table 1 and Table 5.
>
>   Regarding the ablation setup: (1) w/o GT Fusion removes the geometric fusion module, reverting the model to a dual-branch structure akin to EP-BEV, where causally enhanced features are directly aligned with original street features via contrastive loss, without geometry-aware fusion; (2) w/o CFE eliminates the causal intervention path and its associated supervision. This setting retains the rest of the pipeline but loses the guidance provided by causal enhancement.
>
> ### **Q2.** Regarding the hyperparameters experiment.
>
> **A5:** We thank the reviewer for raising this important concern. As the reviewer suggested, we performed a sensitivity analysis on the weights of the two auxiliary objectives: $\alpha$ for the contrastive loss between street and BEV features before fusion, and $\gamma$ for the contrastive loss between the causally enhanced features and the fused features.
>
> The $\alpha$-weighted loss serves as a pre-alignment regularization mechanism, encouraging the street and BEV features to be geometrically consistent before fusion. While we conducted ablation experiments on different values of $\alpha$, we observe that this parameter has a relatively minor impact on performance. Therefore, due to space constraints, we omit the detailed table. Intuitively, overly large $\alpha$ may over-constrain the feature space and interfere with effective fusion, while small values provide slight regularization benefits without significant influence.
>
> For the $\gamma$ coefficient (**Table Ⅲ**), we conducted controlled experiments and observed that setting $\gamma = 0.5$ yields the best performance across datasets. In the main paper, we conservatively used $\gamma = 0.1$ to avoid potential model collapse that could undermine the street-to-satellite alignment. Although this collapse was not observed during training, we adopted a cautious setting to ensure robustness. We will include the results corresponding to the optimal $\gamma$ value in the final version and explain this design choice accordingly.
>
> **Table Ⅲ. CVACT_val-C-ALL | Effect of Varying γ (causal supervision weight, α fixed at 0.1)**
>
> |           | R@1       | R@5       | R@10      | R@1%      |
> | --------- | --------- | --------- | --------- | --------- |
> | 0.1 (now) | 88.68     | 95.58     | 96.66     | 98.49     |
> | 0.3       | 88.92     | 95.78     | 96.84     | 98.46     |
> | 0.5       | **89.49** | **95.84** | **96.92** | 98.49     |
> | 0.7       | 89.25     | 95.83     | 96.90     | **98.63** |
>
> ### **Q3.** Regarding the visualization.
>
> **A6:** We thank the reviewer for the helpful comment and apologize for the limited qualitative visualizations in the main text. While examples are included in the appendix, we will move key results into the main paper for clarity. As shown in **Figure 5** , the baseline model tends to focus on visually salient but semantically irrelevant regions (e.g., sky, background), revealing sensitivity to spurious correlations. In contrast, our model, guided by geometric topology and causal learning, consistently attends to meaningful structures such as road layouts and vegetation, capturing more stable and view-invariant features while effectively filtering out confounding noise.
>
> ### **Q4.** Comparing with recent baseline.
>
> **A7:** We thank the reviewer for emphasizing the need for broader baseline comparisons and appreciate the suggested references. However, several of these works had not released official code or pretrained models at submission time, making timely reproduction difficult. Moreover, some focus on different tasks (e.g., language-based or sequential localization), which are not directly comparable to our single-image setting.
>
> To address this concern, we conducted additional experiments comparing our method against two strong and peer-reviewed baselines that are publicly available and more aligned with our setting:
>
> - GeoDTR+ [TAPMI 2024].
> - Cross-View Geo-Localization with Street-View and VHR Satellite Imagery in Decentrality Settings [arXiv 2025].
>
> We evaluated these baselines on CVUSA and CVACT_val using their official implementations. As shown in **Table Ⅳ** (to be added in the final version), our method consistently outperforms them, demonstrating its effectiveness and generalizability.
>
> **Table Ⅳ.** Comparison with recent and representative baselines on three public benchmarks: CVUSA and CVACT_val.
>
> |         | CVUSA     |           |           |           | CVACT_val |           |           |           |
> | ------- | --------- | --------- | --------- | --------- | --------- | --------- | --------- | --------- |
> |         | R@1       | R@5       | R@10      | R@1%      | R@1       | R@5       | R@10      | R@1%      |
> | Ours    | **98.73** | **99.71** | **99.80** | **99.84** | **91.61** | **96.93** | **97.72** | **98.77** |
> | DReSS   | 98.45     | 99.61     | 99.72     | 99.83     | 91.35     | 96.67     | 97.36     | 98.62     |
> | GeoDTR+ | 95.05     | 98.42     | 98.92     | 99.77     | 87.76     | 95.50     | 96.5      | 98.32     |

---

> ### Comment · Area_Chair_CWJ1 · 2025-08-05
> **Discussion**
>
> Dear Reviewer W79E,
> Please have a look at the rebuttal. Did the rebuttal address your queries.
>
> Could you kindly clarify which of your comments you feel have not been adequately addressed? Additionally, could you specify the areas where the paper still requires significant revision?
>
> I encourage both the reviewers and the authors to actively engage in the discussion to help move the process forward constructively.
>
> Best regards,

---

> ### Author Response · Authors · 2025-08-06
> **Appreciation for Constructive Feedback and Score Adjustment**
>
> Dear Reviewer W79E,
>
> We sincerely thank you for the time and effort you spent reviewing our work and for the constructive suggestions that have helped us improve the paper. In response to your feedback, we will include additional qualitative comparisons on corrupted datasets in the final version to clearly demonstrate the superior performance of our method over existing baselines under challenging conditions.
>
> We would also like to clarify that we have already discussed the limitations and future directions in the conclusion section. Specifically, while our fusion module robustly incorporates BEV geometric information into street features, the model's performance gain is limited when the BEV input is severely degraded. As future work, we plan to explore a more comprehensive causal learning framework tailored to this task to enhance generalization further.
>
> We also intend to release our code and checkpoints to facilitate further research and community engagement. Thank you again for your insightful comments, and we wish you all the best.

---

### Note · Authors · 2025-08-12

Dear AC and Reviewers,

During the rebuttal period, we have addressed most of the reviewers’ concerns. Here, we further detail our causal learning mechanism for reducing confounding factor interference and spurious correlations in CVGL. Following prior causal modeling works and considering the characteristics of street images, we first establish an SCM for CVGL, then perform causal intervention guided by this SCM.

SCM Overview
As illustrated in Figure 1, our SCM is grounded on the following assumptions:

- The street image $X_s$ is mainly generated from two sources: semantic content $C$ and a domain confounder $D$ (e.g., background, lighting), denoted as $C \rightarrow X_s \leftarrow D$.
- The content $C$ contains both discriminative and non-discriminative parts. Together with $D$, the non-discriminative part contributes to the generation of non-causal features $F_n$ via $D \rightarrow F_n \leftarrow C$. The CFE module perturbs a portion of these non-causal components.
- The causal features $F_c$ are derived from the discriminative part of $C$ via $C \rightarrow F_c$.
- The full feature representation $Z = \{F_c, F_n\}$ influences the final prediction $Y$ via $Z \rightarrow Y$, where $Y$ is the matching label.

The overall cross-view matching process can be expressed as $X_a \rightarrow f(X) \rightarrow Y \leftarrow f(X) \leftarrow X_s$, where $X_a$ denotes the aerial image. In this context, the SCM formulation $Z \rightarrow Y$ is a causal abstraction of the model’s computational path $X_s \rightarrow f(X) \rightarrow Y$.

Back-door Adjustment and Intervention Path
Here $C$ denotes confounding variables that influence the generation of $X_s$. Based on the SCM, our goal is to weaken their effect along the spurious path $C \rightarrow X_s \rightarrow f(X) \rightarrow Y$. Since some confounders are explicitly defined in the frequency domain (Line 40 of our paper), we design the CFE module based on this insight. The CFE module itself performs the do-operation $\text{do}(X_s := X_s^*)$ to suppress non-causal components, and we implement the back-door adjustment by enforcing a supervision loss between the causally enhanced (obtained from $X_s^\*$) and fused features, encouraging the fused representation to focus on causal components. CFE implementation details are provided in Rebuttal2 Q2.  Supplementary explanation of front-door and back-door adjustments can be found in Rebuttal4 Q1.

We will clarify this explanation further in the final version.

---

### Decision · Program_Chairs · 2025-09-17

**Decision:**

Accept (poster)

**Comment:**

This paper aims to address the cross-view geo-localization task.  Authors propose a Causal Learning and Geometric Topology (CLGT) framework, which consists of two novel architectural contributions - a causal feature extractor to focus on stable and task-relevant features, and GT fusion to efficiently inject the Bird’s Eye View (BEV) feature with road topology. The paper also proposes data-adaptive pooling, which outperforms conventional global max pooling, global average pooling, and adaptive pooling on the task at hand.

The rebuttal successfully addresses several queries of the reviewers related to causal features extractor, normal cross-attention based fusion, data-adaptive pooling, hyper-parameters tuning  and more recent baselines.

As recommended by reviewers, authors are encouraged more qualitative results and thorough error analysis.
AC recommends paper acceptance